



# On the Potential of Aerodynamic Pressure Measurements for Structural Damage Detection

Philip Franz[1], Imad Abdallah[2], Gregory Duthé[2], Julien Deparday[3], Ali Jafarabadi[2,4], Alexander Popp[1,5], Sarah Barber[3], and Eleni Chatzi[2]

[1]German Aerospace Center (DLR), Institute for the Protection of Terrestrial Infrastructures, Rathausallee 12, 53757 Sankt Augustin, Germany
[2]Institute of Structural Engineering, ETH Zürich, Stefano-Franscini-Platz 5, 8093 Zürich, Switzerland
[3]Institute of Energy Technology, Eastern Switzerland University of Applied Sciences, Oberseestrasse 10, 8640 Rapperswil, Switzerland
[4]Empa, Swiss Federal Laboratories for Materials Science and Technology, Überlandstrasse 129, 8600 Dübendorf, Switzerland
[5]University of the Bundeswehr Munich, Institute for Mathematics and Computer-Based Simulation (IMCS), Werner-Heisenberg-Weg 39, 85577 Neubiberg, Germany

**Correspondence:** Eleni Chatzi (chatzi@ibk.baug.ethz.ch)

**Abstract.** This study investigates the potential of using aerodynamic pressure time series measurements to detect structural damage in elastic, aerodynamically loaded structures. Our work is motivated by the increase in the dimensions of modern wind turbine blade designs, whose complex behavior necessitates the adoption of improved simulation and structural monitoring solutions. In refining the tracking of aerodynamic interactions and their effects on such structures, we propose to exploit aero-
5  dynamic pressure measurements, available from a novel, cost-effective and non-intrusive sensing system, for structural damage assessment on wind turbine blades. This study is based on a series of wind tunnel experiments on a NACA 633418 airfoil. The airfoil is mounted on a vertically oscillating cantilever beam with structural damage introduced in form of a crack by gradually sawing the cantilever beam close to its support. The pressure distribution on the airfoil is measured under diverse configurations of inflow conditions and structural states, including different angles of attack, wind velocities, heaving frequencies, and crack
10  lengths. We further propose an algorithm, relying on convolutional neural networks, for damage detection and rating based on the monitored signals. Analysis of the dynamics of the system using reference acceleration measurements and a finite element model and application of the suggested method on the experimental data indicate that aerodynamic pressure measurements on airfoils can indeed be used as an indirect approach for damage detection and severity classification on elastic, beam-like structures in mildly turbulent environments.





# 1 Introduction

The increasing global demand for sustainable energy sources has led to a significant rise in the deployment of wind turbines (WTs) and their development in terms of capacity and size. Between 2020 and 2022, the average annual installations experienced an increase to $88.7$ GW (Global Wind Energy Council, 2023), marking a notable rise of $56\%$ with respect to the period from 2015 to 2019, where this number stood at $56.7$ GW. Both the rotor diameter and hub size of newly installed horizontal axis WTs have continuously increased over the last years, as more swept rotor area amplifies energy capture. In land-based wind energy in the USA, for example, the rotor diameter and hub size increased by $3\%$ and $4\%$, respectively, between 2021 and 2022 to an average of $131.6$m and $98.1$m (Wiser et al., 2023). Offshore wind energy in the USA reflects the same trends (Musial et al., 2023). Along with blade length, blade flexibility has correspondingly increased in modern WTs (Veers et al., 2023). With the structural complexity of WTs thus rising (Brondsted et al., 2023), it is important to devise efficient mechanisms for ensuring the structural integrity as well as for guaranteeing the long-term operational efficiency and reliability of these critical infrastructures. This task is taken on by Structural Health Monitoring (SHM) systems, which aim to exploit diverse sensor measurements to monitor the structural condition of these assets (Avendaño-Valencia et al., 2020; García and Tcherniak, 2019; Chandrasekhar et al., 2021). The rotor of a WT accounts for approximately $20\%$ of the capital expenditures of land-based wind projects (Stehly et al., 2020). Within this assembly, blades have been shown to be particularly susceptible to various damage types, e.g. leading edge erosion, buckling and blade collapse (Mishnaevsky, 2022), with obvious implications to the performance and integrity of the entire turbine. The task of damage identification for WT blades has become more crucial for recent designs, which involve considerably larger rotor blades, which in turn induce higher aerodynamic loading and more complex aeroelastic effects (Veers et al., 2023). To this end, numerous approaches have been proposed for the targeted identification of structural damage on wind turbine blades (Kong et al., 2023; Kaewniam et al., 2022; Ciang et al., 2008), including schemes that rely on the use of vibration- or strain-based monitoring (Ou et al., 2021; Pacheco-Chérrez and Probst, 2022; Laflamme et al., 2016): Laflamme et al. (2016) conduct a numerical case study to propose and demonstrate a method to detect, localize and rank the severity of structural damage on a wind turbine blade (WTB) under wind loads. The suggested method employs measurements from a network of novel strain sensors, relying on the use of a low-cost soft elastomeric capacitor, that are deployed directly on the blade. Pacheco-Chérrez et al. (2023) suggest a multistep procedure based on Operational Modal Analysis (OMA) of acceleration signals, to detect and rank the severity of crack-like damage on rotating WTBs despite the presence of measurement noise. The authors demonstrate the functionality of this method in a numerical study, employing acceleration signals sampled from 30 locations on a wind turbine blade (WTB). Di Lorenzo et al. (2016) propose another OMA-based method to detect damage on WTBs relying on multiple accelerometers directly attached to the blade. They verify their method in a numerical study using acceleration data from six accelerometers and experimentally validate their method on a $6.5$ m long WTB employing eight accelerometers. Weijtjens et al. (2017) elucidate an indirect sensing approach that uses acceleration signals recorded at the substructure of offshore wind turbines for monitoring the structural integrity of the rotor.

Despite the growing need for structural monitoring of WTBs, deploying sensors on blades for industrial applications remains a non-trivial task, since such sensors have to be minimally invasive, wireless and lightweight. To this end, a wireless,





non-intrusive, low-cost micro-electro-mechanical-system (MEMS)-based pressure and acoustic measurement system, termed Aerosense, has been developed by Barber and colleagues (Barber et al., 2022; Polonelli et al., 2023; Deparday et al., 2022; Polonelli et al., 2022). The Aerosense system consists of a sensing node, a base station, and a software pipeline for furnishing an integrated digital twin. The sensing node exploits energy harvesting options for self-sustainability and is outfitted with various sensing modules, including absolute and differential pressure sensors, microphones, and an inertia measurement unit,

all embedded in a flexible sleeve. The sleeve, shown in an experimental setup in Figure 1c) has a thickness of 2.8mm and its length in span-wise direction depends on the blade on which it is deployed. Figure 1a) schematically illustrates the intended

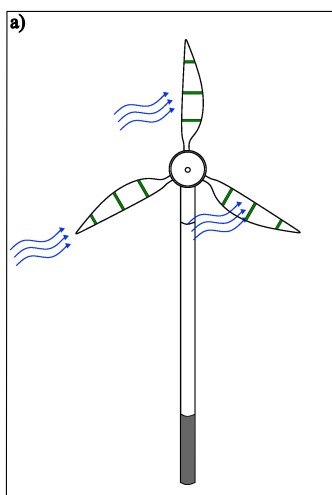
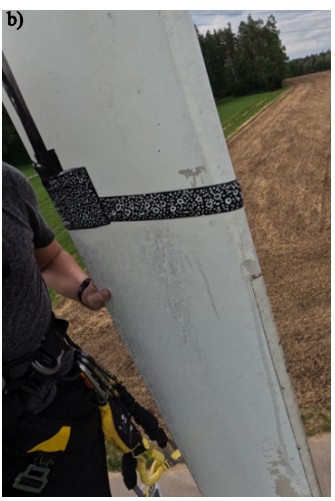
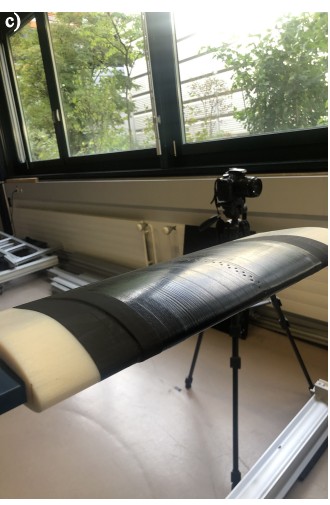

**Figure 1.** a) Intended use of Aerosense sensing units (green stripes) on a wind turbine in a real-world monitoring scenario, designed after Barber et al. (2022); b) Aerosense system deployed on a wind turbine. Image taken from (Abdallah et al., 2023). Panel c): Close-up photo of the Aerosense sensing node (from the experiments conducted for this article).

deployment and use of the Aerosense system in a real-world monitoring scenario with several sensing nodes (green stripes) installed per blade. Figure 1b) illustrates two Aerosense sensor nodes attached to a WTB. Figure 2 illustrates an exemplary pressure distribution recorded with the Aerosense system in the experiments described in Section 3. These recordings are trans-

mitted via Bluetooth to the receiver base station, where the data is then fed to the software pipeline for supporting performance and condition assessment, and digital twinning tasks. The Aerosense system was designed with the main aim of serving for inference of critical inflow quantities, such as the local angle of attack, as well as for the use in WTB leading edge erosion detection. The capacity to infer inflow conditions has been successfully tested and experimentally validated in wind tunnel experiments (Marykovskiy et al., 2023), which corroborated the accuracy and precision of the Aerosense system (Polonelli

et al., 2022). To what concerns the latter aim, a method to detect and classify leading edge erosion has been designed based on synthetic data (Duthé et al., 2021; Barber et al., 2022). Although this system was not designed to serve the purpose of structural damage assessment, the premise of this work is that the indirect measurements offered by the Aerosense node can be





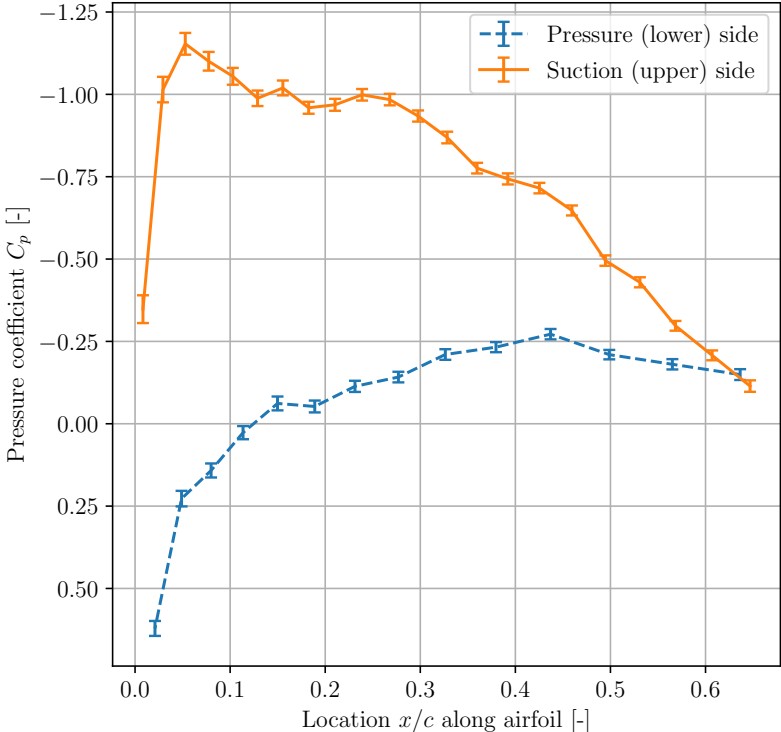

**Figure 2.** Mean value of pressure distribution time series over NACA 633418 with $8°$ angle of attack, a wind velocity of $v = 24\mathrm{m/s}$, an excitation frequency $f_h$ of 1.9Hz and a crack length of 0% of the beam width. Error bars indicate the standard deviation. The pressure distribution was recorded using the Aerosense system in the experiments described in Section 3.

leveraged to this end. With "structural condition assessment" we refer to structural damage detection and the severity rating of the detected damage.

Only few instances in the existing literature attempt to derive structural damage by tracking aerodynamic quantities, with the majority of these efforts focused on aircraft and unmanned aerial vehicles (UAVs). Among these efforts, Zhang et al. (2018) use dimensionless aerodynamic force and moment coefficients as inputs to a fuzzy logic system for fault detection on aircraft. The algorithm relies on inference of aerodynamic force coefficients on the basis of acceleration measurements collected at different positions of the aircraft. Ruangwiset and Suwantragul (2008) utilize aerodynamic lift coefficients for the purpose of

monitoring the structural health of UAVs in a wind tunnel study. The method relies on determining the lift coefficients based on acceleration measurements. As a common denominator of these works, vibration-based measurements are typically exploited, as these are more straightforwardly linked to damage. In this work, we wish to examine the indirect use of aerodynamic measurements, typically serving different purposes, for the task of structural condition assessment. Also related to our work, but not in relation to damage detection, are the following numerical and experimental studies focusing on the aerodynamic

assessment of heaving airfoils. Veilleux (2014) performed numerical studies to analyze the influence of the plunging- and





pitching-, -stiffness and -damping on the oscillation of an aerodynamically loaded and elastically mounted airfoil, with the goal of optimizing fully-passive flapping-airfoil turbines. Ajalli et al. (2007) and Abdi et al. (2008) conducted wind tunnel studies with oscillating airfoils to experimentally investigate the influence of heaving and pitching amplitudes of the airfoil on the sectional aerodynamic pressure distribution around the airfoil. Finally, Madsen et al. (2022) developed the so-called "pressure belt" system, whose purpose and design is similar to the Aerosense solution. The pressure belt system is also packed in a thin sleeve, fixed non-intrusively to a WTB and aimed to measure the pressure distribution and inflow conditions on full scale WTs. However, to our knowledge, the pressure belt system has not been used for damage detection so far.

As highlighted by the sparse literature, detecting structural damage on elastic, aerodynamically loaded structures like WTBs, based on the sectional aerodynamic pressure distribution around an airfoil, seems to be a novel approach. Consequently, we investigate in the present work for the first time whether structural damage, e.g. cracking, can be detected via the measured sectional aerodynamic pressure distribution over a 2D airfoil on an elastic, aerodynamically loaded, and vertically oscillating structure. For this purpose, we conduct a wind tunnel study where we record the aerodynamic pressure distribution over a heaving[1] airfoil under various angles of attack, wind velocities, excitation frequencies and structural states. Damage is introduced to the setup as a crack by gradually sawing the beam close to its support. Subsequently, we propose a multivariate algorithm based on convolutional neural networks (CNNs) to detect and rate the severity of structural damage. The paper is structured as follows. In Section 2, we offer an overview of the adopted hypothesis and its theoretical foundations and associated limitations. Section 3 outlines the adopted experimental setup. The algorithm for damage detection and severity ranking, and its basis, i.e., CNNs, are presented in Section 4. In Section 5, we describe the split of the experimental data and evaluate the proposed damage detection and severity ranking algorithm. Finally, in Section 6, we discuss the results obtained in the previous section based on an the dynamics of the cantilever beam inferred from reference acceleration measurements and a finite element model (FE-model). A summary of main conclusions and directions for future research is offered the in Section 7.

## 2 Method and limitations

This work examines the hypothesis that aerodynamic pressure time series measurements can be exploited, when available, as indirect indicators of structural performance, thus serving the purpose of monitoring both aerodynamic performance and structural condition. This hypothesis is based on the argumentation that structural damage on an elastic structure, e.g. a crack or a delamination in a WTB, will induce a change in the stiffness ($\Delta K$) and damping ($\Delta C$) properties of the WTB. These changes alter the vibration of the elastic structure. Since the aerodynamic pressure distribution (see Figure 2) depends on the interaction of the structure with the surrounding fluid flow, we expect the pressure distribution to be affected by a shift in the vibrational behavior of the blade. Hence, we postulate that changes in the aerodynamic pressure distribution, and correspondingly the normalized pressure coefficient, denoted by $c_p(x,c)$ (see Equation 8), can be leveraged for structural damage assessment. However, the tracking of the evolution of the pressure distribution over time is non-trivial, and typically requires an extensive

---

[1]With a "heaving airfoil", we refer to a vertically oscillating airfoil.





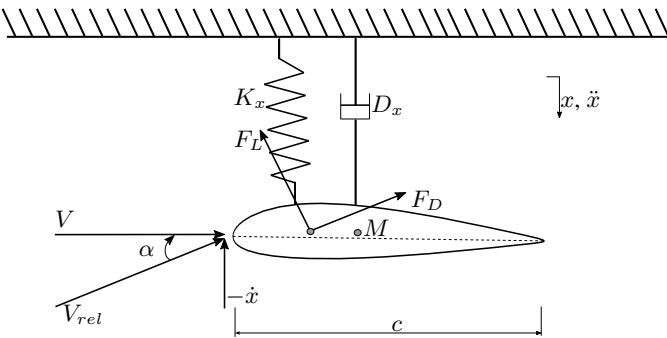

**Figure 3.** Simplified schematic of a rigid, elastically-mounted airfoil with symbolic representation of key parameters. The mass of the airfoil is denoted by $M$, the structural damping by $D_x$ and the spring stiffness $K_x$. The displacement of the airfoil is described by $x$, the velocity by $\dot{x}$ and the acceleration by $\ddot{x}$. The wind velocity is denoted by $V$ and the relative wind velocity by $V_{rel}$. The angle of attack is described by $\alpha$, the aerodynamic lift force by $F_L$ and the aerodynamic drag force by $F_D$.

measurement setup, involving intricate tubing (Traub and Cooper, 2008; Hu and Yang, 2008). We circumvent this issue by adopting the recently developed Aerosense system (Barber et al., 2022) to record the pressure distribution time series. This

approach implicitly captures the variations $\Delta c_p(x, c)$ in the pressure distribution time series, providing an indirect indicator of structural damage. In support of this argumentation, we present a theoretical framework underpinning our hypothesis in Section 2.1 and refer to relevant experimental studies from existing literature in Section 2.2. Subsequently, Sections 2.3 and 2.4 describe an approach to testing the proposed hypothesis and elaborate on associated limitations.

### 2.1 Theory supporting hypothesis

The following set of simplified equations describes the dynamic instability of an airfoil, modeled as a single-degree-of-freedom (SDOF) system incorporating mass and damping, which is subjected to wind inflow and free heaving (see Figure 3). Extensive theoretical exploration has been undertaken to understand the dynamic instability of this system in the context of galloping studies, e.g. in Blevins (1990) or in Modarres-Sadeghi (2021). Equation 1 denotes the equation of motion of the SDOF-system:

$$M\ddot{x} + D_x\dot{x} + K_x x = F_x \tag{1}$$

where $M$ is the mass of the airfoil, while $D_x$ denotes the structural damping and $K_x$ the stiffness in $x$-direction. The displacement, velocity and acceleration of the SDOF airfoil model are denoted by $x$, $\dot{x}$, and $\ddot{x}$. For $F_x$, the resulting vertical aerodynamic force acting on the airfoil, we assume:

$$F_x = \frac{1}{2}\rho V^2 c C_x \tag{2}$$

where $\rho$ is the fluid density, $c$ is the chord length of the airfoil section, $V$ is the horizontal inflow velocity and $C_x$ the total aerodynamic vertical force coefficient. Following Modarres-Sadeghi (2021), $F_x$ can be expressed as a function of the angle of





attack (AoA) $\alpha$ and the lift and drag coefficients $C_L$ and $C_D$ (Equation 3 and 4):

$$\alpha = tan^{-1}\left(\frac{\dot{x}}{V}\right) \tag{3}$$

$$F_x = -\frac{1}{2}\rho c\left(C_L\cos\alpha + C_D\sin\alpha\right)V_{rel}^2 \tag{4}$$

For small angles of attack it follows that:

$$\alpha \approx \frac{\dot{x}}{V} \tag{5}$$

$$F_x \approx -\frac{1}{2}\rho c\left(C_L\cos\alpha + C_D\sin\alpha\right)V^2 \tag{6}$$

Following Blevins (1990), employing a truncated Taylor expansion at $\alpha = 0°$ to linearize Equation 6 and reformulating Equation 1 yields:

$$M\ddot{x} + D_x\dot{x} + K_x x = -\frac{1}{2}\rho V^2 c C_L\big|_{\alpha=0} -\dot{x}\left(\frac{1}{2}\rho V c\left(\frac{\partial C_L}{\partial\alpha}+C_D\right)\bigg|_{\alpha=0}\right) \tag{7}$$

From the right side of Equation 7 follows that a change in $K_x$, $D_x$ or $M$, possibly caused by structural damage, not only affects the vibration of the airfoil, described by $x$, $\dot{x}$ and $\ddot{x}$, but also alters its loading, the vertical aerodynamic force $F_x$. Since a change in $F_x$ implies a change in the pressure distribution $c_p(x,c)$, we conclude that structural damage may indeed affect the pressure distribution around an oscillating airfoil respectively its variation $\Delta c_p(x,c)$. Although these equations indicate a

relation between structural damage and the aerodynamic pressure distribution, one must keep in mind that these equations are simplified; a direct connection between damage and the pressure distribution is not established, as only integrated quantities are accounted for. Additionally, unsteady aerodynamic effects are not taken into consideration in these equations.

### 2.2 Experimental work supporting hypothesis

Our hypothesis is further supported by existing experimental studies. Ajalli et al. (2007) conducted experiments to investigate

the aerodynamic pressure distribution of a heaving airfoil in conditions similar to those of our experimental campaign (see Section 3). These showed that increasing the heaving amplitude of the airfoil leads to amplified absolute values of the pressure coefficient $c_p$ and an amplified lag between the equivalent AoA $\alpha_{eq}$ and the $c_p$ recorded close to the leading edge, which gives hint to an increased aerodynamic unsteadiness. Furthermore, Abdi et al. (2008) suggest that pitching motions with different rotational amplitudes also lead to pronounced changes in the maximal values of the $c_p$ close to the leading edge. As structural

damage changes the modal properties of a structure and consequently its vibrations, these experiments support our hypothesis.

### 2.3 Testing of our hypothesis

The following considerations have led to the experimental setup that we propose in Section 3.





- Rotatory-wing aerodynamics are more complex than fixed-wing aerodynamics. Since the aim of this study is to investigate whether it is at all possible to detect structural damage from aerodynamic pressure, we opt for a fixed-wing experiment. We approximate a fixed WTB by mounting an airfoil on a cantilever beam and placing it in a wind tunnel test section.

- The flap-wise vibration of a rotating WTB is strongly affected by gravitational forces (Diken and Asiri, 2021). For that reason, we introduce periodic forcing at the tip of the beam to emulate the flap-wise vibration of a rotating WTB caused by the gravitational forces. Furthermore, we aim for a heaving amplitude of 5cm to 10cm at the mid-section of the airfoil, to obtain a non-stationary aerodynamic pressure distribution affected by the heaving, and choose the tip mass as well as the excitation frequency accordingly.

- To introduce structural damage, we decrease the cantilever's cross-section by sawing it close to its support. This reduction of the cross-section mimics the stiffness reduction caused by a crack. The crack length will be increased stepwise such that different structural states can be regarded. Within each state, the structure will be subjected to different combinations of AoA, wind loading, and tip excitation.

## 2.4 Scope and limitations of the chosen approach

Our investigation is conducted in a wind tunnel facility under controlled environmental and operational conditions. It should thus be clearly stated that this paper does not aim to answer whether it is possible to detect and rank the severity of structural damage under real operational (rotating wing aerodynamics, pitching, tension stiffening, etc.) and environmental conditions (high turbulence) of a wind turbine. It rather aims to offer a proof of concept as to whether such highly indirect pressure measurements can be conceived for use within an SHM setting. Furthermore, the tasks of damage localization and classification in SHM are not examined herein, since the distribution of Aerosense patches along a blade, i.e., the aspect of sensor placement, is not investigated; instead, we focus on investigations at the airfoil level. Moreover, the periodic excitation employed in our experimental setup may pose a limit for damage identification, as the corresponding periodic vibration response is larger than the transient vibrations caused by the aerodynamic loading and thus could complicate the identification task. The purpose of this work is to instead offer a first indication as to whether aerodynamic pressure distribution time series of low turbulent aerodynamics may be used to detect and rank the severity of damage on elastic and aerodynamically loaded beam-like structures.

## 3 Wind tunnel experiments

## 3.1 Experimental setup

The experimental setup (Figure 4) consists of an airfoil mounted on a flexible cantilever beam, placed in an open test section of a wind tunnel of the Unsteady Flow Diagnostics Laboratory at the École Polytechnique Fédérale de Lausanne (EPFL). The test section measures approximately 40 cm x 40 cm and the wind tunnel can operate with a maximum wind speed of



approximately $35\ \mathrm{m/s}$. As presented in Figure 4, the aluminum cantilever beam has a rectangular cross-section with a height of $1\ \mathrm{cm}$ and a width of $4\ \mathrm{cm}$. The clamped support of the beam is realized via a screwed connection, which is fixed to an aluminum frame, as illustrated in Figure 5 a). The aluminum frame, acting as a support structure, is placed on polymer mats to reduce the influence of ambient vibrations. Close to the support, damage is induced by gradually sawing the beam, which locally reduces its stiffness. The damage severity is quantified by measuring the length of the introduced cut. Multiple "crack" lengths are tested: $0\%$, $12.5\%$, $25\%$, $37.5\%$ and $50\%$ of the overall beam width (see Figure 5 b)). At the mid-span of the beam, the NACA 633418 airfoil with a chord length $c = 16\mathrm{cm}$ and a width of $45\mathrm{cm}$ is aligned with the test section of the wind tunnel. Two 3D-printed airfoils were designed with specific slot angles, where the beam passes through, allowing for two angles of attack (AoA) to be tested, namely $0°$ and of $8°$. As the cantilever beam gets damaged during the tests and changing the AoA would imply removing the airfoil and wide parts of the measurement equipment, we use two identical aluminum beam, one for experiments with $0°$ AoA and one for $8°$ AoA. At the tip of the aluminum beam, a motor, controlled by an analog power source, rotates an eccentric mass and, thus, applies a harmonic excitation, which induces flap-wise bending, as indicated by the red arrows in Figure 5 a). Torsional motions (purple arrow in Figure 5 a)) might also appear under such an excitation, but are secondary relative to the main bending of the blade. The Aerosense system records the sectional aerodynamic pressure distribution over the airfoil profile at the mid-section of the airfoil (see Figure 4). Reference accelerometers are used to record the acceleration in the y-direction at five different positions, as indicated via the orange dot annotation along the cantilever beam shown in Figure 4. The cantilever with the airfoil and the harmonic loading is conceived as a proxy setup, whose purpose is to emulate a rotating WTB oscillating in its flap-wise direction due to gravitational loading.

### 3.2 Measurement system

To record the aerodynamic pressure at the mid-section of the airfoil, we install the Aerosense sensing node, introduced in Section 1, at this position. The sensing node comprises 40 Micro-Electro-Mechanical Systems (MEMS) absolute pressure sensors sampled at $100\ \mathrm{Hz}$, embedded in the previously described sleeve. On this experimental setup, the Aerosense system also has the advantage to be easily installed and reused for the two tested airfoils. Figure 2 shows the pressure distribution measured by the absolute pressure sensors after calibration. To compare pressure distributions from different inflow velocities, we compute the pressure coefficient $c_p(x,c)$ from the Aerosense data using the following expression:

$$c_p(x,c) = \frac{p(x,c) - p_\infty}{\frac{1}{2}\rho_\infty V_\infty^2}, \tag{8}$$

where $c$ denotes the chord length and $p(x,c)$ the pressure measured at the position $(x,c)$ on the pressure or suction side of the airfoil. The pressure measured in the free stream is described by $p_\infty$, the density of air by $\rho_\infty$ and $V_\infty$ is the fluid velocity in the free stream. We compute $p_\infty$ based on the zeroing measurements conducted before and after every test block (explained in the following Subsection 3.3) and $V_\infty$ is set to the wind velocity of the corresponding experiment. Reference piezoelectric accelerometers are installed at five positions along the beam (see Figure 4) and are connected via adhesive petro wax to the surface of the beam. They are positioned at the midsection of the beam in the transverse direction, which is denoted as z-direction in Figure 4. The acceleration signal is sampled at 2000Hz.





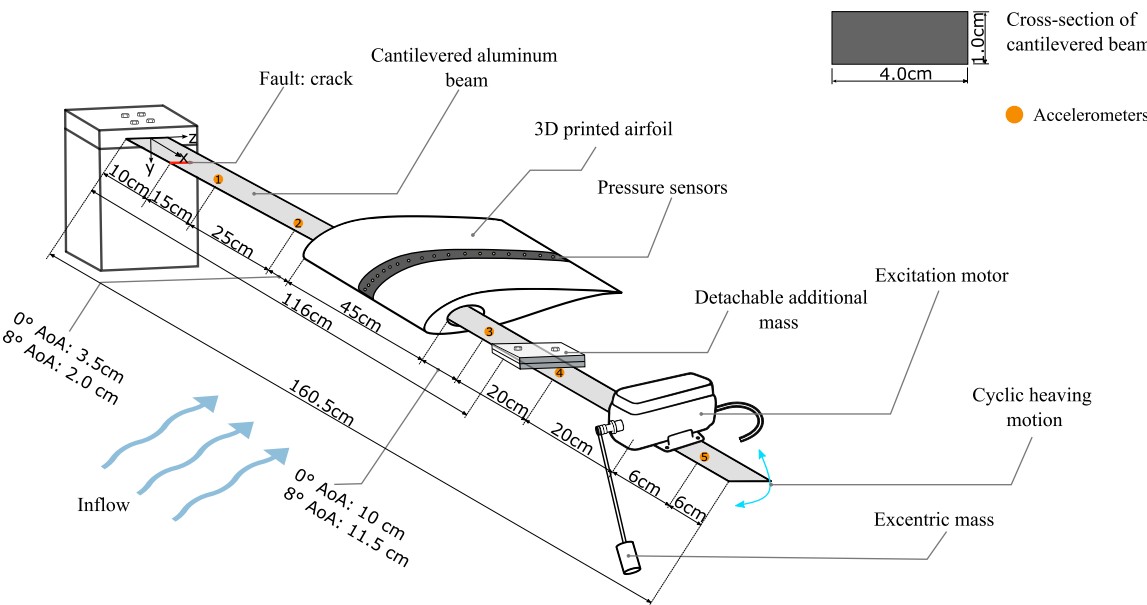

**Figure 4.** The experimental setup at EPFL consists of a cantilever beam carrying an airfoil, an excitation motor and a detachable additional mass and is positioned in the open test section of a wind tunnel. Measurement instrumentation is present on the airfoil in form of the Aerosense system and on the cantilever beam in form of accelerometers. The width and height of the rectangular cross-section of the beam is shown in the right upper corner of the figure.

## 3.3 Design of experiments

During the experiments, the aerodynamic pressure distribution time series at the location of the Aerosense sensing node is recorded for increasing damage severity, induced by the stepwise extended crack length, under various boundary conditions.

Furthermore, we measure the acceleration acting on the beam at the locations of the accelerometers (see Figure 4) in each experiment. To emulate different operational and environmental conditions, the experiments are conducted under two different AoA $\alpha_1 = 0°$ and $\alpha_2 = 8°$. These two angles correspond to two conditions where the derivative of the aerodynamic force with respect to the AoA differs $\left(\left(\frac{\partial C_L}{\partial \alpha}\right)\right.$ and alters the aerodynamic damping in Equation 7). For each AoA, two different wind speeds ($V_1 = 12$ m/s, $V_2 = 24$ m/s) and two different excitation frequencies ($f_{h,1} = 1.0 Hz$, $f_{h,2} = 1.9 Hz$) are considered.

The Reynolds number is respectively $Re = 1.28 \times 10^5$ and $Re = 2.56 \times 10^5$ and the reduced frequency $k$ lies in the range of $0.021$ to $0.08$. The Reynolds number and reduced frequency were computed using the following expressions, where $\mu_\infty$ denotes the free-stream kinematic viscosity of air and $\rho_\infty$ the free-stream density of air:

$$Re = \frac{\rho_\infty V c}{\mu_\infty}, \quad \text{with} \quad \mu_\infty = 1.7894 \times 10^{-5} \frac{kg}{ms} \quad \text{and} \quad \rho_\infty = 1.2250 \frac{kg}{m^3} \quad \text{(Anderson (2017))} \tag{9}$$

$$k = \frac{\omega_M c}{2V}, \quad \text{with} \quad \omega_M = 2\pi f_h \quad \text{in} \quad \left[\frac{rad}{s}\right] \quad \text{(Hodges and Pierce (2011))} \tag{10}$$



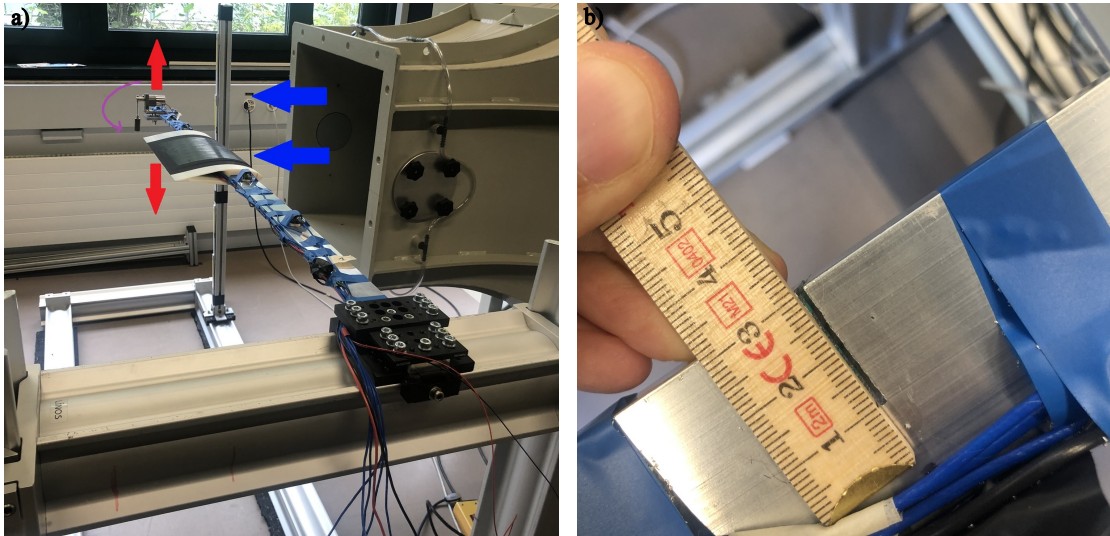

**Figure 5.** Panel a): Experimental setup. The blue arrows indicate the direction of wind flow from the wind tunnel, the red arrows show the direction of oscillation of the cantilever beam and the purple arrow points to the direction in which the beam twists. Panel b): an example of the sawed cantilever beam, where the crack length is 20mm. View from above.

We use the standard values for $\mu_\infty$ and $\rho_\infty$ at sea-level altitude and 15°C, as these approximate the environmental conditions during the experiments. Two AoAs, two wind speeds and two heaving frequencies result in eight test series (TS) that are summarized in Table 1. For each test series, we consider five different crack lengths as well as one state with an added mass and a crack length of $0$ mm. Thus, there is a total of six structural states per test series (see Table 3). For each combination of structural state, AoA, wind velocity and heaving frequency, we conduct three measurement runs (later also called 'experiments') of approximately 150s. During the first 15s, the airfoil is only aerodynamically loaded and for the remaining 135 s aerodynamic and harmonic forces both act on the airfoil simultaneously. After each "test block" consisting of a set of three experiments, we conduct zeroing measurements and re-fasten the screws on the support structure to ensure that the structural boundary conditions remain identical for all experiments. As during the zeroing measurements the cantilever beam is only loaded by ambient vibrations, these acceleration recordings of these measurements will be used later for structural identification. Every measurement respectively experiment is identified by its unique number; an overview over the numbering of experiments is given in the Tables A1 (harmonic and wind loading), A2 (only harmonic loading) and Table A4 (ambient loading) and the file "design of experiments" in the supplementary material. The same experimental numbering is applied for both AoAs.





# 4   Data Analysis

In this section, we propose a method to detect and rank the severity of structural damage employing the experimentally acquired
pressure distribution time series. Our suggested method is based on a machine learning approach which relies on the use of a
CNN. In what follows, we therefore first offer a brief introduction to CNNs and then explain the details of our method.

## 4.1   Convolutional neural networks

CNNs are a subclass of artificial deep neural networks that have been successfully applied as supervised learning schemes
across a broad variety of tasks, such as image classification, natural language processing, and time series classification (LeCun
et al., 2015; Ismail Fawaz et al., 2019). A CNN comprises convolutional layers, pooling layers, and fully connected layers
(LeCun et al., 2015), which are schematically shown in Figure 6. Applying a convolutional layer on input data (see Figure
6) essentially translates into applying multiple filters on the input data (LeCun et al., 2015). However, in contrast to normal
filters, the weights of convolutional layers are not predefined, but learnable. One or several convolutional layers are usually
followed by a pooling layer. Pooling layers, like "local" or "global", "average" or "max" pooling, reduce the dimensions of the
previously extracted convoluted feature maps by taking the average or maximum of a window with a certain size (local pooling)
or of the whole feature map (global pooling) (Ismail Fawaz et al., 2019). The last, fully connected layer(s) of a CNN determine
for the output of the preceding pooling layer the probability distribution over the regarded classes (Ismail Fawaz et al., 2019).
Thus, this last layer classifies the features, which were extracted beforehand by the convolutional layers and condensed by the
pooling layers. While the activation function "ReLU" is often used in convolutional layers, the activation function "softmax" is
used in the last layer for multi-class classification. During the training phase in a supervised learning scheme, the filter values
(weights and biases) of the convolutional layers are optimized through backpropagation. Thus, convolutional layers learn to
extract discriminative features (Ismail Fawaz et al., 2019).

## 4.2   Proposed damage detection and severity ranking algorithm

The proposed damage detection and severity rating algorithm is based on the CNN architecture proposed by Wang et al.
(2017). This architecture, shown in Figure 7, consists of three convolutional layers, where each one is followed by a batch
normalization layer and activated with the ReLU activation function, a global average pooling (GAP) layer and a final, fully

**Table 1.** Boundary conditions of the eight test series.

| Test series | 1 | 2 | 3 | 4 | 5 | 6 | 7 | 8 |
|---|---|---|---|---|---|---|---|---|
| AoA [°] | 0 | 0 | 0 | 0 | 8 | 8 | 8 | 8 |
| $f_h\ [Hz]$ | 1.0 | 1.0 | 1.9 | 1.9 | 1.0 | 1.0 | 1.9 | 1.9 |
| $v\ \left[\frac{m}{s}\right]$ | 12 | 24 | 12 | 24 | 12 | 24 | 12 | 24 |





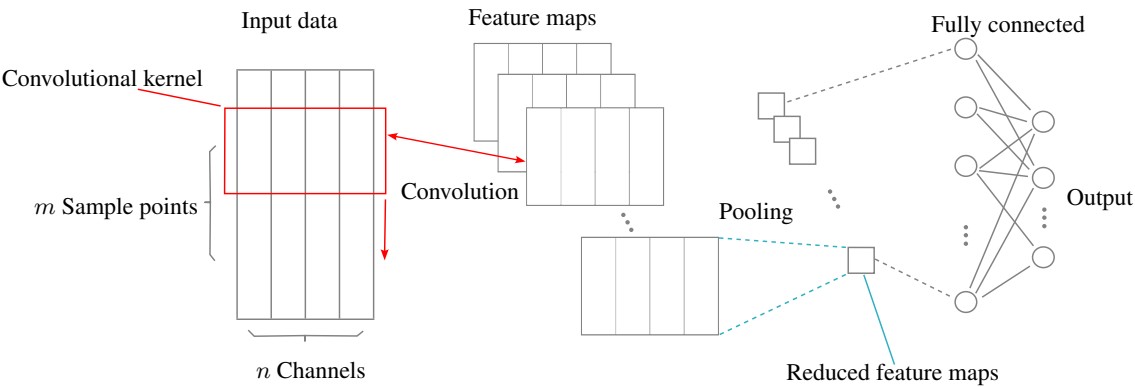

**Figure 6.** Simplified schematics of a CNN architecture that takes multivariate time series windows with a length of $m$ sample points and $n$ channels as input. A 1D convolution is carried out by a "sliding" a convolutional kernel vertically over the time series data. The feature maps resulting from convolution with different kernels are condensed by a pooling layer. The classification of the compressed feature maps, coming from the pooling layers, takes place in the fully connected layers at the end of the network.

connected layer using the softmax activation function. Our input data consists of multivariate time series samples. On these time series samples, the first convolutional layer applies $128$ kernels with a length of $8$, the second one $256$ kernels with a length of $5$ and the third one $128$ kernels with a length of $3$. All convolutional layers use zero padding, a stride of $1$ and are

followed by batch normalization layers to accelerate the convergence and to improve the generalization of the network (Wang et al., 2017). Subsequently, in the GAP layer, the feature maps coming from the third convolutional layer are strongly reduced and classified by the final, fully connected layer.

We decided to adopt a CNN-based architecture instead of a more sophisticated network architecture, as for example recurrent or attention based neural networks that were also successfully employed for time series classification (Ismail Fawaz et al.,

2019; Mohammadi Foumani et al., 2024). Our decision is motivated by the fact that the number of trainable parameters for CNNs is much smaller than for the mentioned alternative architectures. As the size of our experimental dataset is small, we expect a CNN to better generalize due to fewer learnable parameters and the more sophisticated architectures to overfit on our training data set. We initialize the weights and biases of the network using Glorot's uniform initialization scheme (Glorot and Bengio, 2010) with a fixed random seed and employ the Adam algorithm (Kingma and Ba, 2015) to optimize the network

weights and biases to minimize the loss function (sparse categorical cross-entropy loss). Following Ismail Fawaz et al. (2019), we use a model checkpoint procedure on the validation set, meaning that the model that performs the best on the validation set during the training process is used for the final evaluation. Furthermore, we use a batch size of $10$ multivariate time series samples and we initiate the training process with a learning rate of $0.05$ that we reduce by a factor of $0.5$ once the validation loss of the model does not improve for $15$ epochs. The learning rate reduction is also done according to Ismail Fawaz et al.



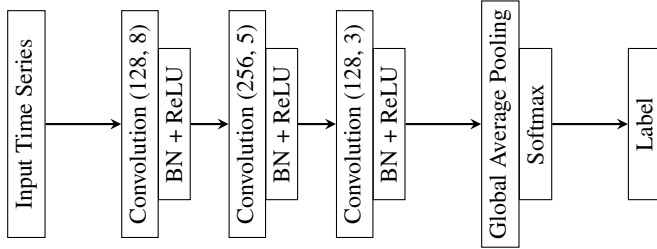

**Figure 7.** CNN architecture for time series classification proposed by Wang et al. (2017) consisting of three convolutional layers with batch normalization and rectified linear unit activation functions (BN+ReLU), and a GAP-layer with a softmax activation function. The convolution layers have either 128 or 256 kernels with a size of 8,5 or 3 - as indicated by the number in the respective field. The figure is based on Figure 1 of Wang et al. (2017).

(2019); however, as we train our CNN for 150 epochs only, we increase the starting value of the learning rate to 0.05 and decrease the number of epochs required for an update of the learning rate from 50 to 15 to accelerate the optimization of the network weights. Apart from that we do not modify any hyperparameters of the CNN proposed by Wang et al. (2017).

The input to the neural network consists of multivariate time series windows with a length of 1.5s. Figure 9 illustrates in panel a) an extract of 15s length of the raw signal recorded by sensor 14 (located at $\frac{x}{c} = 0.02$ on the suction side of the airfoil) of

the Aerosense system under 0° AoA, $V = 12$m/s, and $f_h = 1.0$Hz (experiment 3, see Table A1). In the spectrogram in panel b) of Figure 9 and in the power spectral density (PSD) in panel c) of the same Figure, one can see that the maximum power of the signal is concentrated around approximately 1.95Hz and that this is also the only distinctive peak in the signal. This also applies beyond 16Hz. We observe the peak at $\approx 1.95$Hz and the absence of further distinctive peaks also for the other sensors and for measurements conducted with 8° AoA (see Figure A1). The spectrogram in panel b) is based on a discrete short-term-

Fourier-transformation, conducted with Scipy (Virtanen et al., 2020). The PSD in panel c) is computed using Welch's method and its implementation in Scipy (Virtanen et al., 2020). Both, the spectrogram and the PSD, use the raw pressure time series recorded by sensor 14 in experiment 3 with 0° AoA. The peak in the PSD at approximately 1.95Hz closely coincides with the eigenfrequency of the first flap-wise eigenmode of the cantilever beam (see Table A4). Therefore, we choose signal windows of 1.5s length as input for the CNN, such that each time window contains approximately three periods of pressure variations

related to the vibration of the cantilever beam. As one can further see in the spectrogram shown in spectrogram in panel b) of Figure 9 and Figure A1, there are low-frequent disturbances present in the frequency band between 0Hz - 0.8Hz. These disturbances are most likely caused by sensor drift. By employing signal windows with 1.5s length as input data, we also seek to limit the influence of these disturbances on the classification results. After extracting these signal windows from the raw pressure time series, we normalize each multivariate signal window by deducting the mean of the window and dividing it by

the overall standard deviation of the signal window. Together with splitting the experimental data into a training, validation and test set, this leads to the damage detection and ranking algorithm presented in Figure 8. We chose to only minimally tune the algorithmic hyperparameters as we would like to show that structural damage assessment based on aerodynamic pressure measurements can be achieved with a generic ML scheme, without necessitating much effort. It is not our goal to achieve an





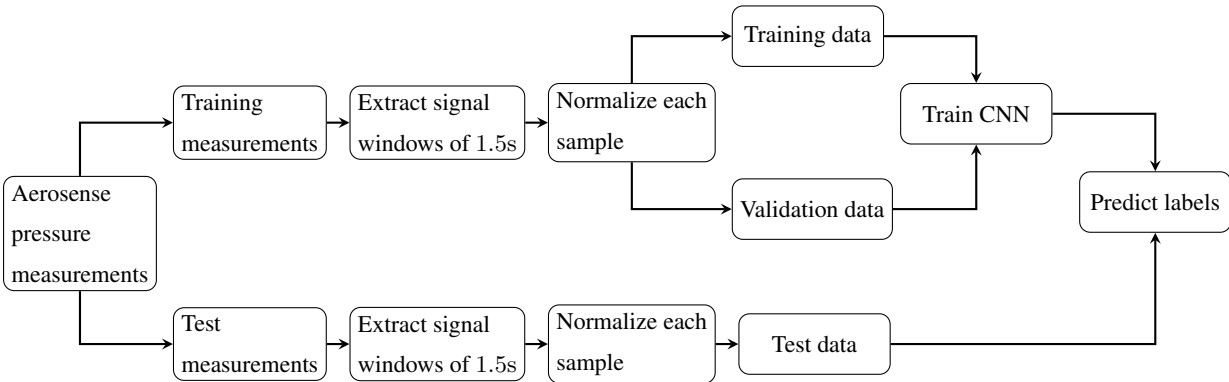

**Figure 8.** Steps of the proposed multivariate damage detection and severity ranking algorithm.

optimized classification accuracy on such a small dataset, since this might not generalize well beyond our experimental setup. Rather, we aim to ensure that replication of our results may be easily achieved. For our implementation we employed Keras 2.14.0 (Chollet et al., 2015) with the TensorFlow 2.14.0 (Abadi et al., 2015) back end.

## 5 Results

This section first explains how the experimental data is divided into training and test data sets. Then, the accuracy of the proposed method for damage detection and severity classification is evaluated on these training and test data sets.

### 5.1 Split of the experimental data

For detecting and ranking structural damage in the acquired experimental data with the method proposed in Section 4, we create two datasets: one that contains all measurements with $0°$ AoA and one with $8°$ AoA, as the pressure distribution strongly depends on the AoA. As described in Section 3.3, for every combination of inflow conditions and structural states we record three sequential measurements of approximately $150$s duration. As we regarded two heaving frequencies $f_h$, two wind velocities $V$ and six structural states, in total 72 multivariate pressure time series measurements are available per AoA. The three measurements of each combination of inflow conditions and structural states are not randomly divided into the basis for the training and test set; instead, a certain 'split' uses always the same two measurements of the three available measurements of every combination of boundary conditions to generate training data, reserving the remaining measurement to generate test data. Thus we can evaluate the performance of our method on completely unseen data. Table A1 in the appendix gives an overview over the boundary conditions of each experiment. Referring to the columns of Table A1, Table 2 describes the composition of the three splits. Consequently, we employ $48$ experiments to generate the training samples and $24$ experiments for the test samples. From each experiment, we neglect the first $40$s and the last $10$s to reduce the influence of transients from switching on the harmonic excitation and from possibly switching off the harmonic excitation too early. From the remaining approximately $100$s we extract 89 multivariate signal windows of $1.5$s with an overlap of approximately $30\%$. As a result, there are for



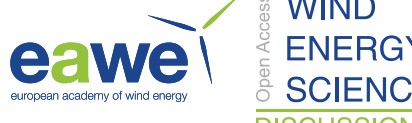

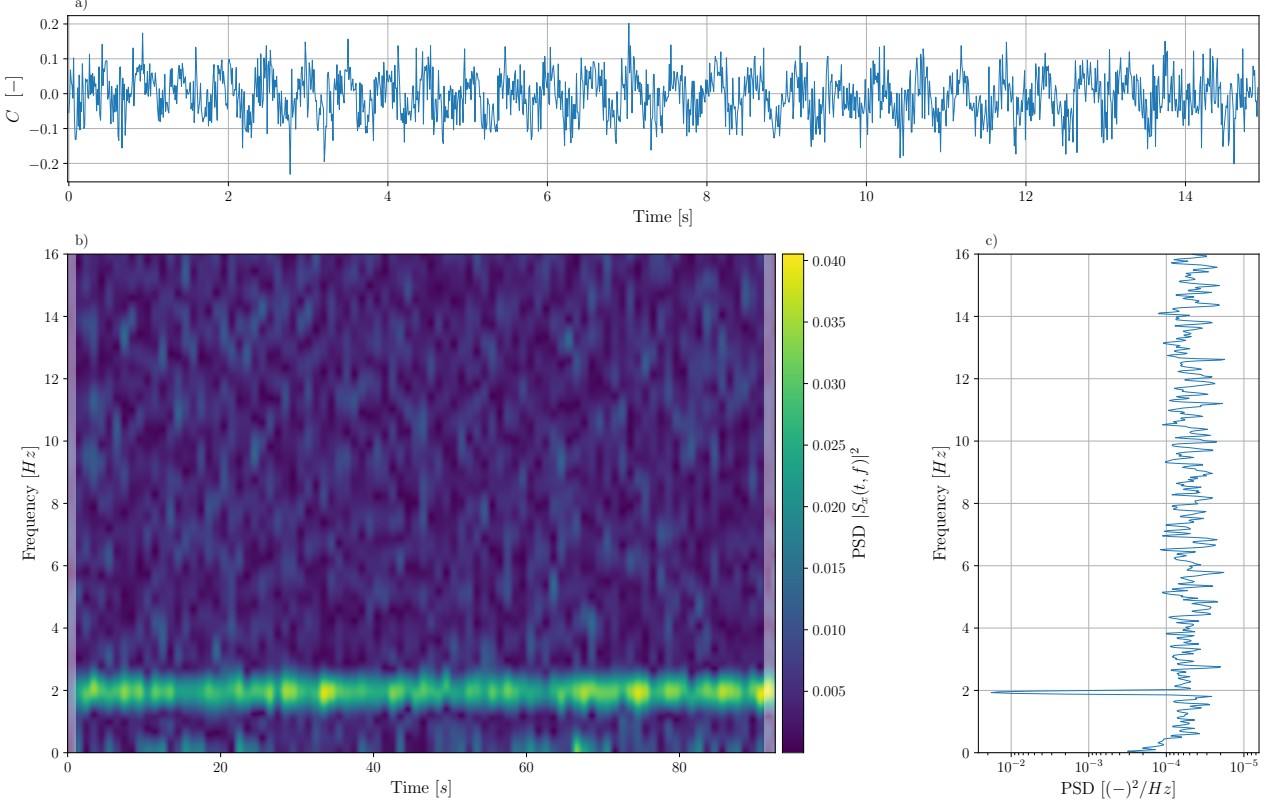

**Figure 9.** a) Extract of the pressure time history recorded by sensor 14, located at $\frac{x}{c} = 0.02$ on the suction side of the airfoil, with $0°$ AoA under $V = 12\frac{m}{s}$ and $f_h = 1.0$Hz in experiment 3. Panel b) shows how the spectrum of the pressure time history of sensor 14 of experiment 3 with $0°$ AoA varies over the whole duration of the experiment via a spectrogram. The spectrogram is computed using the discrete short-term-Fourier-transformation of Scipy (Virtanen et al., 2020). The shaded areas at the left and right ends of the spectrum indicate areas affected by sliding windows that are partially outside of the analyzed signal. Panel c) depicts the PSD of the same signal computed over the whole duration of experiment 3 and is calculated with Welch's method and its implementation in Scipy (Virtanen et al., 2020).

**Table 2.** Composition of the splits used to create the training and test sets to evaluate the multivariate algorithm. The "columns" refer to the columns 1, 2 and 3 of Table A1.

| Split | Experiments for test set | Experiments for training set |
|---|---|---|
| 1 | Column 1 | Column 2 ∪ Column 3 |
| 2 | Column 2 | Column 1 ∪ Column 3 |
| 3 | Column 3 | Column 1 ∪ Column 2 |




training in total 4272 samples available and for testing 2136 (see Table 3). Finally, 25% of the training samples are reserved
for validation purposes and only the remaining 75% of the training samples are de facto used for training the CNN. Thus, per
damage class there are 534 samples available for training, 178 samples for validation and 356 samples for testing, as presented
in Table 3.

**Table 3.** Available samples for training, validation and testing per class.

| Crack length [% of beam width] | 0 | 0 + added mass | 12.5 | 25 | 37.5 | 50 | sum |
|---|---|---|---|---|---|---|---|
| Class number | 0 | 1 | 2 | 3 | 4 | 5 | - |
| Number of training samples | 534 | 534 | 534 | 534 | 534 | 534 | 3204 |
| Number of validation samples | 178 | 178 | 178 | 178 | 178 | 178 | 1068 |
| Number of testing samples | 356 | 356 | 356 | 356 | 356 | 356 | 2136 |

## 5.2 Accuracy of damage detection and severity ranking based on aerodynamic pressure measurements

The results of the proposed multivariate damage detection and severity ranking algorithm are presented subsequently via
confusion matrices in Figure 10. We consider in our experiments structural states with a crack length of 0%, 12.5%, 25%,
37.5% and 50% of the beam-width, as well as one state with 0% crack length and an additional mass of 246g attached to the
beam (see Table 3). The mapping of the structural states to the class numbers can be found in the first row of Table 3.
Figure 10 a) depicts the classification results for split 2 of the data set with $0°$ AoA, and Figure 10 b) for $8°$ AoA.

For the data with $0°$ AoA, the classification of the aerodynamic pressure times series samples works very reliably. Only two
samples from a damaged state are classified as undamaged and these even stem from the state with the smallest damage (see
Figure 10 a)). For $8°$ AoA, the classification works also well, but 12% of the samples of damage class 5 are falsely classified
as damage class 4 (see Figure 10 b)). However, these two classes are neighboring and the detection of a damage from any
of these two states would require an immediate response. Figure 10 shows the classification accuracy of the best performing
split. In Table 4 we present the the classification accuracy averaged over all balanced classes for all regarded splits. The overall
slightly lower classification accuracy for $8°$ AoA and the corresponding more diffuse confusion matrices are likely to be related
to the more turbulent, unsteady aerodynamics which typically emerge due to more detached flows at higher angles of attack.
Furthermore, the systematically lower classification accuracy for first split of both datasets is remarkable. We hypothesize that
the systematically lower classification accuracy for the first split (see column one of Table 4) is caused by uncaptured transients
in the wind flow resulting from wind tunnel activation, as the wind tunnel was the only device in the experimental setup running
all the time during the three subsequent experiments conducted for each combination of boundary conditions. This is a point
that must be investigated and improved in future experiments. However, as the averaged classification accuracy lies for both
datasets at 91.6% and 89.2%, respectively, (compare Table 4) and the lower bound of all classification results is at 80.0%,
our experiments confirm our hypothesis that structural damage can be consistently detected and ranked based on aerodynamic





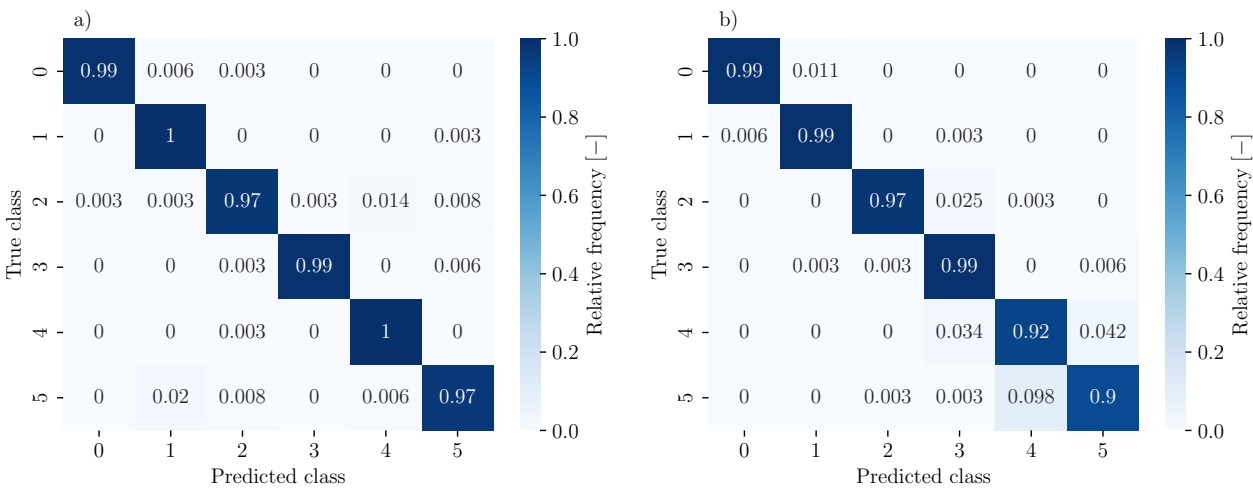

**Figure 10.** Classification results for split 2 depicted by confusion matrices. The columns correspond to the true class of a sample; the rows to the class predicted by the proposed method. The relative frequencies are rounded to three decimals places. a) shows the results for $0°$ and b) $8°$ AoA.

**Table 4.** Accuracy for all splits for $0°$ AoA and $8°$ AoA. The classification accuracy given for a single split is the average of the classification accuracy over the six balanced classes of that split.

| AoA | Split 1 | Split 2 | Split 3 | Average |
|-----|---------|---------|---------|---------|
| $0°$ | 82.6% | 98.5% | 93.7% | 91.6% |
| $8°$ | 81.8% | 96.0% | 89.8% | 89.2% |

360   pressure measurements in mildly turbulent environments. Considering further that the experimental data comprises different Reynolds regimes and excitation frequencies, these results indicate that the proposed damage detection and ranking pipeline based on a indirect sensing scheme is robust towards moderate variability in the environmental and operational conditions and suited for damage detection on elastic, beam-like structures in mildly turbulent environments and warrants further investigation. Since our proposed time series classification method is purely data-driven, we cannot provide more extensive reasoning for how

365   varying the AoA affects detectability. From a computational perspective our algorithm is efficient: once the model is trained, classifying the whole test set is done in $500\,\mathrm{ms}$ on a desktop computer with an Intel i9-10980XE CPU. This demonstrates that our proposed machine learning pipeline is well suited for structural health monitoring in the scope of real-time digital twin applications. Moreover, CNNs can be easily parallelized on GPUs or other ML specific hardware, which may potentially allow for fast on-the-edge damage detection for wind turbine blades when integrated with a system like the Aerosense node.



## 6 Discussion: analysis of system dynamics based on acceleration measurements

In this section, we use the reference acceleration measurements (see Section 3.3) to investigate the intricacy of the damage detection and classification task we attempt to solve with our proposed approach, as introduced in Section 4. Using the hypothesis of Section 2, the preceding section discusses how to adopt a CNN-based scheme to detect structural damage from aerodynamic pressure measurements in a supervised manner. An alternative approach to damage detection would be to rely on vibration-based information. By analyzing the measurements of the installed acceleration sensors, we can more targetedly investigate the dynamic behavior of the system that was previously only indirectly observed via the aerodynamic pressure measurements. For that purpose, we analyze subsequently how the amplitudes of the acceleration signal at the tip of the cantilever beam, and the eigenfrequencies $f_i$ of excited eigenmodes $\phi_i$ develop with increasing crack length. Finally, we also discuss the influence of the orientation of the crack on the dynamics of the cantilever beam using a Finite Element model of the cantilever beam.

### 6.1 Evolution of the vibration amplitudes with increasing damage

The choice of examining the acceleration amplitude as a proxy to damage is motivated by two facts: Firstly, the cantilever beam in the experimental setup is expected to vibrate in a quasi-steady state after initial transients have been damped out, under influence of the periodic forces $F_\Omega$ caused by the rotating mass and the aerodynamic loading $F_{aero}$. We here assume that the system oscillates in a quasi-steady state, since $F_\Omega$ and the lift and drag forces of the airfoil vary periodically and the motion of the blade may additionally induce small vertical, transient loads caused by vortex shedding. As a result of the quasi-steady vibration state, a reduction of the beam stiffness is expected to lead to amplified vibration amplitudes under the prescribed loading. The more usual features to detect damage based on output-only acceleration data, such as eigenfrequencies, may be non-trivial to determine under forced conditions (when the input force is not measured). Secondly, the choice of analyzing the evolution of the acceleration amplitude is motivated by its linkage to pressure. Based on Equation 7, the velocity $\dot{x}(t)$, and thus also the displacement $x(t)$ and acceleration $\ddot{x}(t)$ of the airfoil, serve as inputs to the aerodynamic pressure distribution and its variation. To better understand the variation of the pressure distribution, we analyze subsequently the evolution of its inputs that are related to the dynamics of the cantilever beam by examining the acceleration amplitude with growing damage. Since the beam vibrates in a quasi-steady state, the evolution of the displacement and velocity amplitudes with increasing damage can be approximately inferred from the evolution of acceleration amplitudes.

The acceleration signal recorded at the tip of the cantilever beam comprises oscillations at several different frequency ranges, as observed from the multiple peaks present in the PSD plot in Figure 12. The PSD in Figure 12 is computed using the implementation of Welch's method (Welch, 1967) in MATLAB (2021), after the acceleration signal is filtered with a band-pass filter with a pass-band between $1.0\,\mathrm{Hz}$ and $60\,\mathrm{Hz}$ and sampled-down to $200\,\mathrm{Hz}$. To analyze the evolution of the acceleration amplitudes with increasing damage, we compute the root mean square (RMS) value of the acceleration time histories, as this feature characterizes the average amplitude of a noisy signal. The RMS of a discrete time signal $y_k$ with $k \in [0, 1, ..., K-1]$ is computed as $rms_y = \sqrt{\frac{1}{n} \sum_{k=0}^{K-1} y_k^2}$. For computing the RMS, we neglect the first $40\,\mathrm{s}$ and the last $10\,\mathrm{s}$ of each acceleration signal to limit the influence of initial transients or shutting down the rotational motor too early.



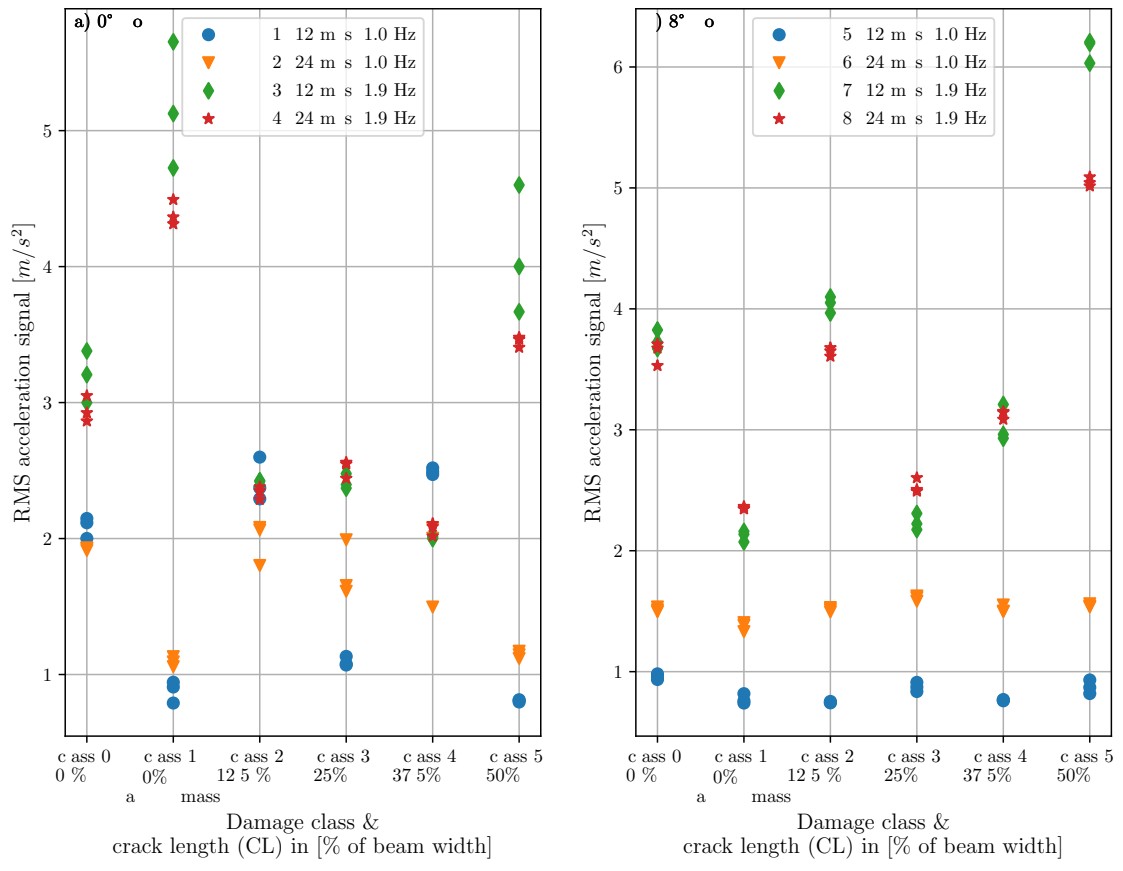

**Figure 11.** RMS values of the acceleration signals recorded by accelerometer 5, located at the tip of the cantilever beam, for all test series (TS) and all damage classes. Each data point marks the RMS value of a single acceleration measurement. The color and marker of the data point indicate the corresponding TS and thus the approximate loading conditions. Panel a) shows the RMS values of experiments conducted with $0°$ AoA. Panel b) displays the RMS of acceleration measurements recorded with $8°$ AoA. Since for each combination of inflow conditions and structural state, three measurements were conducted, three circles are visible per TS/damage class combination.



Figures 11 a) and b) report the RMS of the accelerations measured with sensor 5 at the tip of the cantilever beam for all TS and every damage class. Every data point marks the RMS value of an acceleration signal in a single experiment conducted under the rotor, $F_\Omega$, and aerodynamic, $F_{aero}$, loads. The RMS value is computed as previously described. In panel a), we observe the RMS recorded for $0°$ AoA, while in panel b) results are reported for $8°$ AoA. The color of the data points refers to the TS of the measurement and indicates the loading conditions of the system during the measurement. Looking at the evolution of the RMS for a fixed set of loading conditions over different damage classes, e.g. TS 1 in panel a), it is observed that despite a growing crack and a, thus, increasingly reduced stiffness of the cantilever beam, the recorded acceleration amplitudes do not increase monotonically. Figure 11 does not reveal a strictly monotonic increase in acceleration RMS values with increasing crack length in all TS. We hypothesize that the non-monotonic trends result from the complex loading conditions and the orientation of the introduced crack and investigate this further in the subsequent sections. Furthermore, the vibration amplitudes that occur for different damage classes often overlap. This hold true, for instance, for the vibration amplitudes observed at crack lengths corresponding to 12.5% and 25% of the beam width for both examined AoAs. This implies that the use of vibration-based measurements as damage proxies would not be straightforward in this case, and further reveal that the classification task which we solve using the CNN-based scheme is a non-trivial one.

## 6.2 Evolution of the eigenfrequencies with increasing damage

As a further damage proxy, we here conduct an approximate calculation of the modal frequencies (eigenfrequencies) of the beam under increasing damage. We already explained that this calculation is non-trivial under a forced excitation regime, since the Operational Modal Analysis (OMA) requirements are not fulfilled. OMA methods are output-only, that is, only require response measurements (as is the case for the available measurements here) and require that the examined structure is subjected to loadings with white-noise or at least broad band characteristics that excite all mode shapes of interest (Brincker and Ventura, 2015). Therefore, we opt to use the acceleration measurements of all five accelerometers from the zeroing runs (see Section 3.3), where only ambient loading was present on the cantilever beam to conduct the modal analysis. We use covariance-driven subspace identification (SSI), introduced by van Overschee and de Moor (1996), and its implementation in `n4sid()` of MATLAB (2021). The acceleration signals are windowed to have a length of $60s$ for most of the experiments, are preprocessed with a band-pass filter with a pass-band between $1.0Hz$ and $60.0Hz$, and sampled down to $200Hz$. Since the accelerometers installed on the beam can only measure accelerations along the $y$-direction, we can only detect mode shapes which exhibit vertical oscillations. In the following, we only compute the eigenfrequencies and eigenmodes for the cantilever beam with the $0°$ AoA airfoil, as both employed cantilever beams are nearly identical (see Section 3).

To determine stables poles in the stability diagrams, we use a tolerance of $1\%$ deviation between the natural frequencies of two poles from subsequent model orders and a tolerance of $20\%$ deviation of the modal assurance criterion (MAC), introduced by Allemang and Brown (1982), of the eigenmodes of two poles of subsequent model order. Using the resulting stability diagrams (see Figure 12 exemplarily for experiment 15) we determine the range of stable model orders. For each pole with stable frequency and MAC, we then compute, from the set of stable model orders, the mean $\mu_{f,i}$ and standard deviation $\sigma_{f,i}$ of the eigenfrequencies $f_i$. The results of $\mu_{f,i}$ and $\sigma_{f,i}$ for the eigenmodes between $1.0Hz$ and $60Hz$ are given in Table A4.



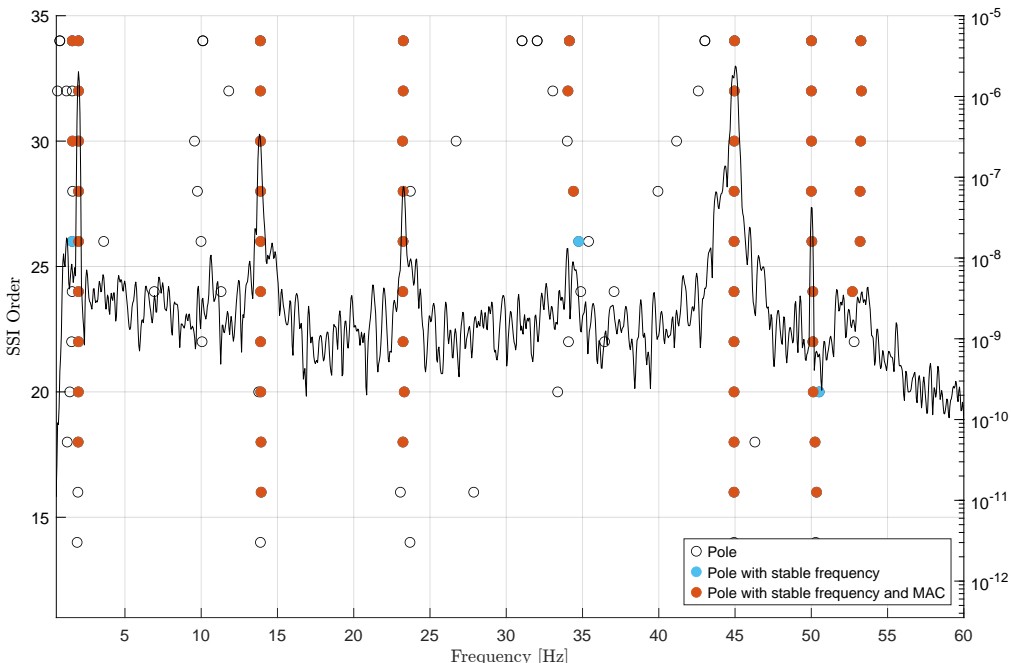

**Figure 12.** Stabilization diagram of experiment 15, where the cantilever beam is undamaged and excited by ambient loading only. The black circles mark the identified frequencies (poles), the blue circles mark poles where the frequency between two subsequent model orders varies less than 1% (stable frequency), while the orange circles mark poles with a stable frequency and where the modal assurance criterion, introduced by Allemang and Brown (1982), between the eigenmodes of two poles of subsequent model order is higher than 80% (stable mode shapes). Furthermore, the PSD of the acceleration signal recorded by sensor 5 in experiment 15 at the tip of the beam, sampled down to 200Hz and filtered as described above, is plotted over the model orders. One can see that the peaks of the PSD and the stable poles align well. The PSD was computed with Welch's method (Welch, 1967) implemented in Matlab (MATLAB, 2021)

The eigenmodes $\phi_i$ that have been identified for experiment 15 are exemplarily given in Figure 14. The evolution of the eigenfrequencies, detected with SSI, for increasing damage is shown in Figure 13.

In Figures 13 a) and b) the eigenfrequencies $f_1$ and $f_2$ of the first two vertically oscillating eigenmodes $\phi_1$ and $\phi_2$ do exhibit
a monotonically decreasing trend with increasing crack length. This seems to hold to some extent also for the evolution of the eigenfrequency $f_6$ of the sixth vertically oscillating eigenmode $\phi_6$, shown in Figure 13 f). However, the sixth eigenmode is not detected in the experiments for damage class 1 and 3, whereas the values for damage class 2 seem to not fulfill the monotonicity pattern. The evolution of the eigenfrequencies $f_3$ and $f_5$ is non-monotonic and for $f_4$ difficult to judge due to a lack of values. Possibly, the eigenmodes corresponding to $f_3$, $f_4$ and $f_5$ are not merely vertically oscillating eigenmodes,
but coupled ones, or sensitive towards small changes in the boundary conditions of the system. To obtain further insight in the these eigenfrequencies and eigenmodes, we offer the results of a simulation in the next section, which aims to replicate the experiment. For the monotonically and approximately monotonically decreasing eigenfrequencies $f_1$, $f_2$ and $f_6$ holds that the



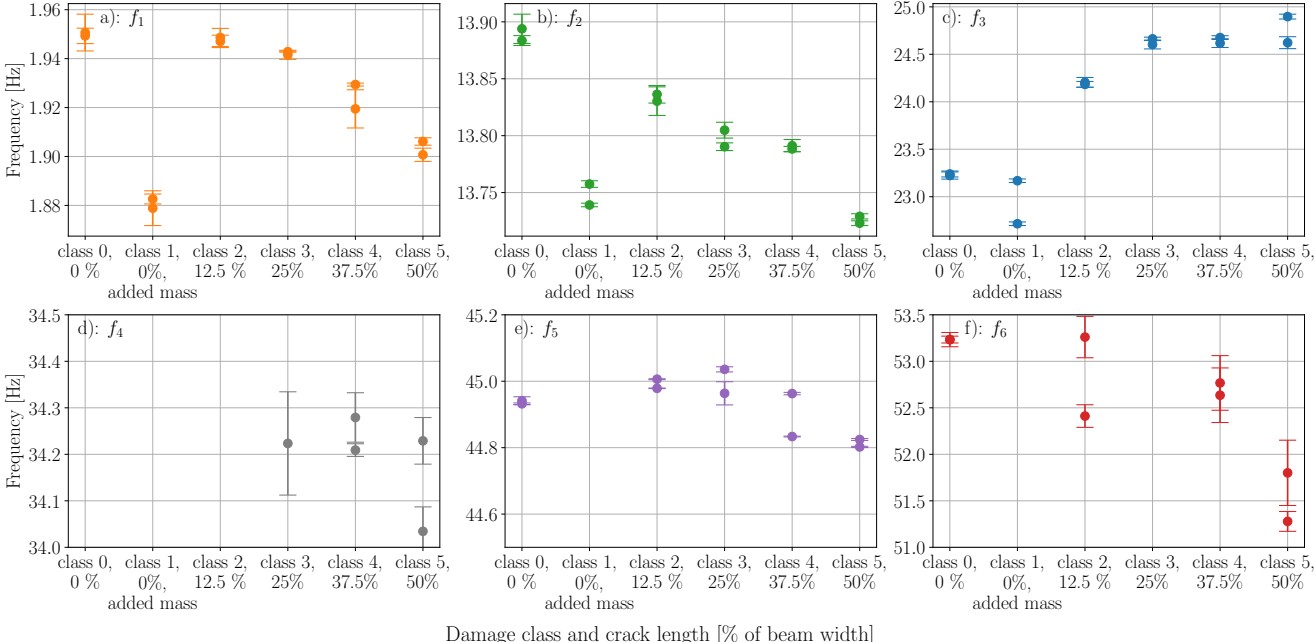

**Figure 13.** Evolution of the mean values (circular markers) and standard deviations (error bars) of the first six eigenfrequencies $f_i$, determined with SSI, plotted over the damage classes. The exact values of $f_i$ for each damage class are given in Table A4. Eigenmode $\phi_4$ is detected only from damage class 3 onward. Hence, no values are shown for damage classes 0, 1 and 2 for $f_4$. The values of $f_5$ in damage class 1 are at approximately 42Hz and therefore not visible. The sixth eigenmode $\phi_6$ and its eigenfrequency $f_6$ could not be detected in the experiments of class 1 and class 3; therefore, there are no values for these classes.

absolute change between the least and the most damaged state is approximately 0.05Hz for $f_1$, 0.16 Hz for $f_2$ and 1.69 Hz for $f_6$.

Based on the above, we deduce that the local stiffness reduction introduced by the crack only bears a limited effect on the eigenfrequencies of the identified eigenmodes, rendering this as well an uncertain proxy for damage detection. We expect the orientation of the crack to contribute to variability of the estimated quantities, since the selected crack orientation is not chosen to reflect a pure bending crack, but rather a defect that allows large vertical oscillations under the chosen wind inflow conditions to affect the aerodynamic pressure distribution on the airfoil. We further investigate this effect based on the aforementioned
simulation model.

## 6.3 Influence of the crack characteristics

In order to gain insights into the experimental configuration and some of the aforementioned observations, we have developed a Finite Element (FE) model of the tested system and report on the estimated evolution of the acceleration amplitudes under constant loading and boundary conditions. In our simulated analysis we neglect the aerodynamic force, $F_{aero}$, since model-

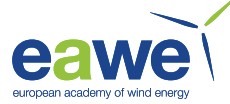


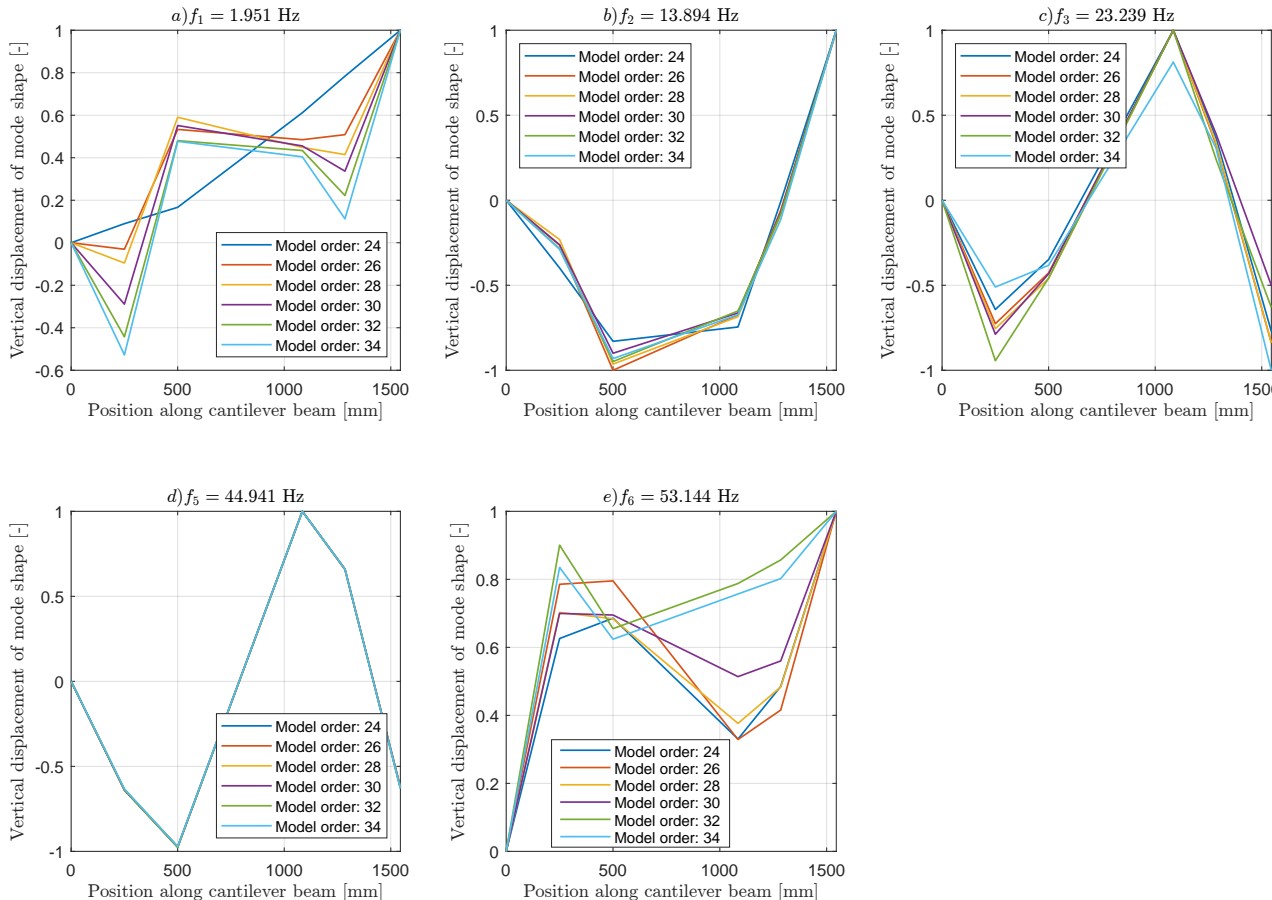

**Figure 14.** Mode shapes of the first five vertically oscillating eigenmodes $\phi_i$ for the stable model orders in experiment 15, determined with SSI and the stability diagram in Figure 12. Eigenmode $\phi_4$ is not detected in experiment 15 (see Table A4). The mode shapes are scaled to comprise a maximum value of 1.0.

ing the dynamic fluid-structure-interaction occurring at the airfoil requires complex treatment, moving beyond the scope of simulating the essential dynamics of this system. The employed FE-model, shown in Figure 16 a), is set up in the Abaqus 2022 finite element package and simulates the response of the cantilever beam under eccentric harmonic loading $F_\Omega$. The cantilever beam is modeled with 8-node general, shear deformable, shell elements with reduced integration (SR8 elements in Abaqus). Furthermore, a rigid beam section using B31 Timoshenko shear-flexible elements was added to the edge of the

cantilever beam at the location of the rotating mass in order to account for the eccentricity of the periodic excitation. The displacement and rotation degrees of freedom at the left end of the beam are set to zero to model the rigid support of the cantilever (encastre boundary condition in Abaqus). The Young's modulus and density of the aluminum beam are set to 70 GPa and $2.66e-06$ kg/mm$^3$, respectively. The periodic resultant force $F_\Omega$ acting on the cantilever can be expressed as a function of the rotating mass ($m_\omega$), eccentricity radius ($r$) and excitation frequency ($f$): $F_\Omega = m_\omega r (2\pi f)^2$ and yields 0.777N





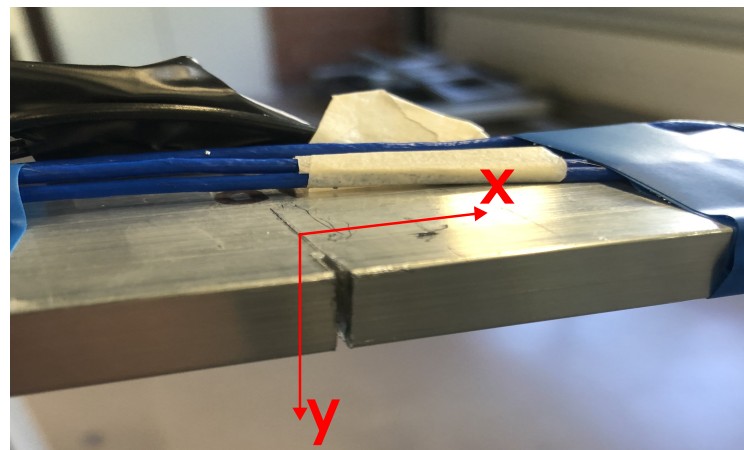

**Figure 15.** Orientation of the actual crack present during the experiments. The rotational and aerodynamic loading lead to vibrations in the vertical respectively $y$-direction of the cantilever beam.

for an excitation frequency of $f_{h,1} = 1.0$Hz. The airfoil and the body of the excitation motor are modeled as lumped masses at the respective positions on the cantilever beam. As a consequence, the model neglects the additional stiffness introduced by the airfoil mounted on the cantilever beam in the experimental setup. The edge crack is introduced into the model as a seam object, resulting in a nodal discontinuity at the crack location with different lengths, ranging from 0mm to 20mm. To determine the first five eigenfrequencies $f_{i,m}$ and eigenmodes $\phi_{i,m}$ of the FE-model, we conduct an eigenvalue analysis using the Lanczos

eigensolver in Abaqus. The resulting eigenfrequencies $f_{i,m}$ and eigenmodes $\phi_{i,m}$ are shown in Figure 16 b) and in the first row of Table A3. To compute the time-dependent response of the cantilever beam under harmonic loading, a dynamic (modal) analysis is carried out based on the results of the eigenvalue analysis. In such a dynamic (modal) analysis, the structural response of the cantilever beam under harmonic loading $F_\Omega$ is determined for an excitation frequency $f_h = 1.0$Hz for a period of 100s and with a time increment of $\Delta t = 0.001$s. A modal damping ratio of 0.03, acting on the first mode, is considered.

For each regarded crack length, such a simulation is conducted. Afterwards, the time history of the vertical acceleration at the location of sensor 5 (close to the tip of the beam, highlighted in Figure 16 and Figure 4) is extracted.

Comparing the eigenfrequencies and mode shapes determined from the experiments (see Figure 14) with those of the FE model (see Figure 16), one can see that the second and fourth mode shapes of the model $\phi_{2,m}$ and $\phi_{4,m}$ are not detected with the OMA and that the third, fourth and fifth eigenmodes of the OMA, $\phi_3$, $\phi_4$, and $\phi_5$ are not found by the eigenvalue analysis

of the FE model. Since the second and fourth mode of the FE-model oscillate only horizontally, these cannot be consistently identified with the experimental acceleration data, as the accelerometers along the cantilever beam only measure vertical accelerations. Furthermore, looking at the mode shape $\phi_3$ and $\phi_5$ (see Figure 14, c) and d)), it is observed that the cantilever beam remains approximately straight between 200 mm and 800 mm. This can be attributed to the additional stiffness which the airfoil adds to the structure. Since the airfoil is only represented as a lumped mass in the FE-model, this model cannot

accurately capture this effect. Mode shape $\phi_4$ only occurs in damage class 3, 4 and 5 and is thus likely a complex eigenmode





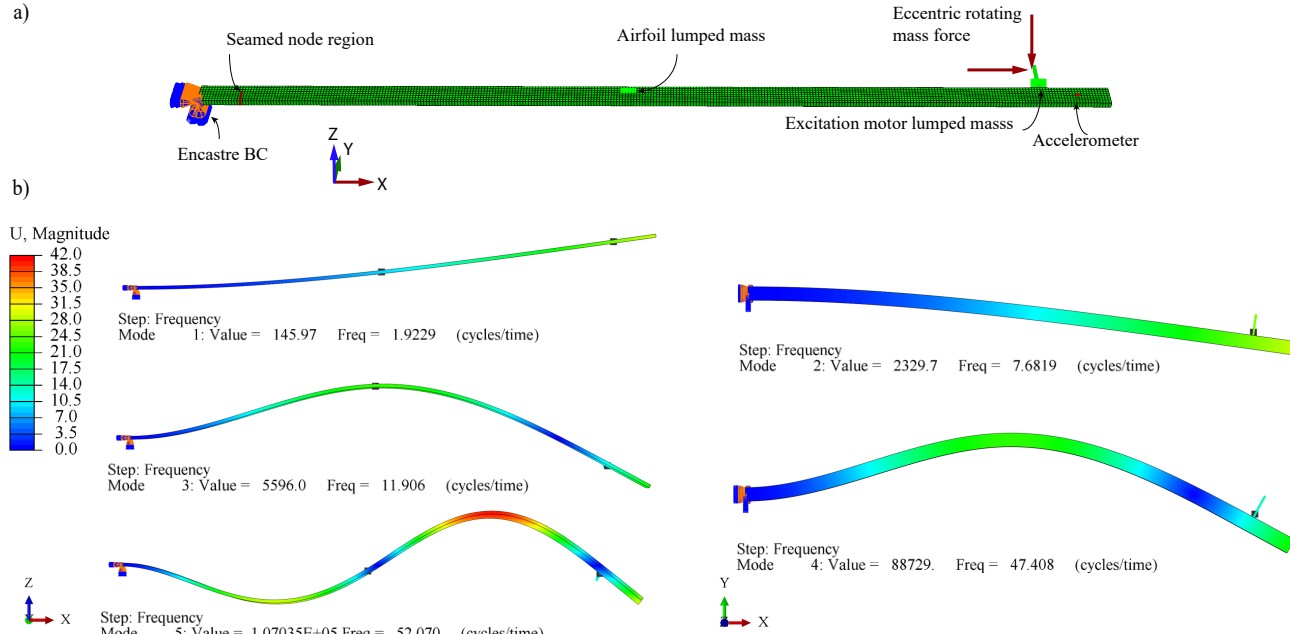

**Figure 16.** a) FE-model of the cantilever beam. The presence of the airfoil and the excitation motor is reflected by the lumped masses. The vertical edge crack is introduced to the model as a seam object, resulting in a discontinuity with different lengths varying from 0mm to 20mm. The effects of the eccentric rotating mass are modeled as harmonic loading with eccentricity in the dynamic modal analysis. b) First five mode shapes $\phi_{i,m}$ and corresponding eigenfrequencies $f_{i,m}$ and eigenvalues of the reference setup (undamaged beam).

related to the crack and therefore not found in the eigenvalue analysis of the FE-model. The modes shapes $\phi_1$, $\phi_2$ and $\phi_6$ inferred via SSI (see Figure 14 a), b) and f)) are similar to the FE-estimated mode shapes $\phi_{1,m}$, $\phi_{3,m}$ and $\phi_{5,m}$ (see Figure 16). Also, their corresponding eigenfrequencies, with $f_1 \approx 1.95$Hz, $f_2 \approx 13.88$Hz, and $f_6 \approx 53.23$Hz, and $f_{1,m} = 1.93$Hz, $f_{3,m} = 11.91$Hz, and $f_{5,m} = 52.070$Hz, are close to each other. Based on this, we conclude that the simulation model offers a good approximation of the experimental setup and may be used for further analysis.

Figure 17 reports the evolution of the RMS of the simulated accelerations (in the $y$-direction), recorded at the position of the sensor 5 of FE-based cantilever beam, on the different damage classes without additional mass. The system is loaded by a harmonic excitation with a frequency of 1.0Hz. In this model, we do not account for the added mass; thus, damage class 1 is missing. In the simulated case, the RMS of the acceleration signal appears to be monotonically increasing with growing crack length. Furthermore, the PSDs of the acceleration signals of sensor 5 of the FE-model for all damage classes without added mass are shown in Figure 18 a). The simulated acceleration signals have a sampling frequency of 1000Hz and the PSD is computed with Welch's method (Welch, 1967) and the Python implementation in Scipy (Virtanen et al., 2020). The panels b), c) and d) of Figure 18 show in detail the peaks in the PSDs corresponding to the eigenfrequencies $f_{1,m}$, $f_{2,m}$, and $f_{5,m}$ of the model. When examining the first two modes, subplots b) and c) reveal that for increasing damage the peaks in the PSDs shift





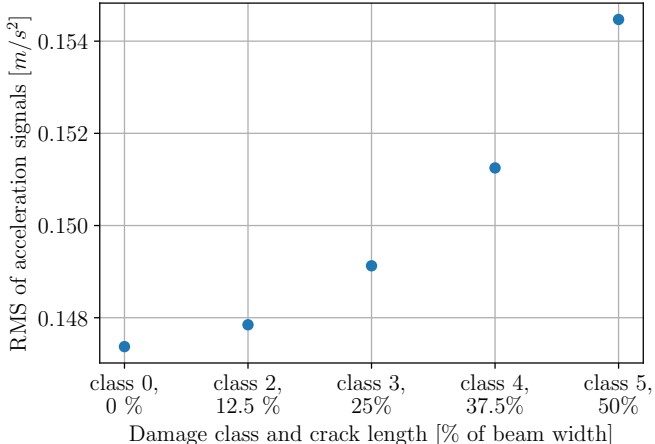

**Figure 17.** RMS of the simulated acceleration response at the location of sensor 5, in the FE-model, for all damage classes without an added mass. The system is loaded by a harmonic force with an excitation frequency of $f_h = 1.0$ Hz.

toward lower frequencies with increasing crack length, meaning that the eigenfrequencies $f_{1,m}$ and $f_{2,m}$ decrease. Moreover, the magnitude (energy) of these peaks seems to uniformly increase, which implies that the vibration amplitudes of these eigenmodes are also increased. For the third vertically oscillating eigenmode in d), the eigenfrequency also appears to decrease with increasing damage, but unlike the behavior for the first two modes, the magnitude of the peaks does not increase. This last observation implies that the RMS is likely not a concise damage proxy. The model does hint that a (monotonic) decrease

in eigenfrequencies is expected for this type of crack, and, indeed, such a decrease is roughly observed in the experimental data. In the experimental forced tests, which aimed to mimic response under wind inflow conditions (Figure 11), we observe more pronounced variability in the response amplitude, and in the variation of the observed frequencies. This can be due to variations in the loading conditions, small variations in the support conditions throughout the experiment, noise (disturbances) due to vibrating sensor cables along the beam, as well as owing to the measurement noise present in the experimental setup.

We expect especially variations in the excitation frequency to contribute to this variability, since firstly, we could set these only approximately in the experimental setup (see Section 3.3) and secondly, the second excitation frequency $f_h = 1.9$ Hz is quite close to the first natural frequency of the setup and causes resonance effects in the vibrations. The discrepancies exhibited during the experiment are also expected to appear on site under operational conditions. As the resulting damage detection and rating problem is non-trivial, we we refrain from using standard unsupervised vibration-based methods as the reference in this

case. To tackle the added complexity of utilizing indirect pressure measurements, rather than vibrational (structural) responses, we employ the proposed supervised learning approach. This approach relies on a learning algorithm designed to account for the latent information embedded in the signals during these experiments.



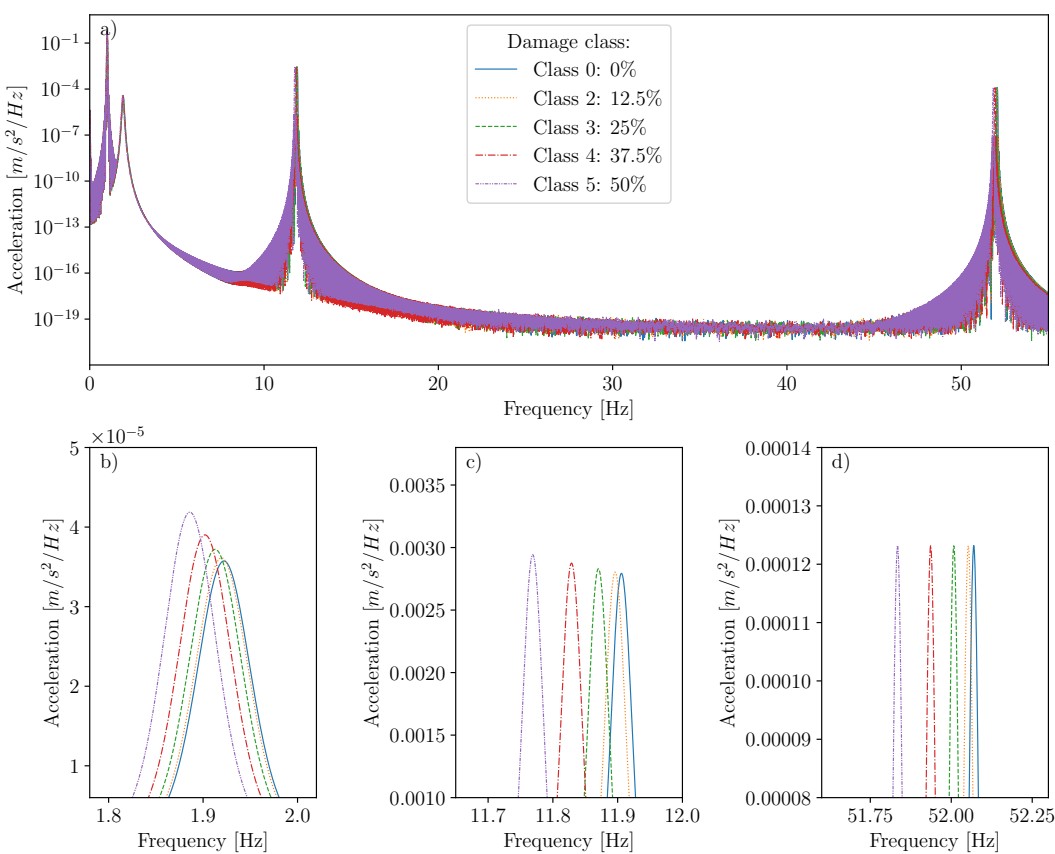

**Figure 18.** Panel a) displays the PSDs of the acceleration signals extracted at the position of sensor 5 from the simulated structural response of the FE-model of all damage states without an additional mass. The simulated signals of sensor 5 are sampled with 1000Hz. The PSD is calculated with Welch's method (Welch, 1967) and its Python implementation in Scipy (Virtanen et al., 2020). Panel b) shows the evolution of the peak of $f_{1,m}$ in detail on a linear scale, and panels c) and d) of $f_{2,m}$ and $f_{5,m}$ respectively. The legend shown in panel a) is valid for panels b), c) and d) as well.



# 7 Conclusions

In this work, we show for the first time that it is possible to detect and rank the severity of structural damage on an elastic,
aerodynamically loaded, beam-like structure based solely on measurements of the sectional aerodynamic pressure distribution
over a 2D airfoil. To demonstrate this, we conduct a wind tunnel study where a NACA 633418 airfoil is mounted on a heaving
cantilever beam. We record the sectional aerodynamic pressure distribution over the airfoil under various boundary conditions,
comprising different angles of attack, wind velocities, heaving frequencies and damage states, using the Aerosense system, a
cost-effective and non-intrusive, MEMS-based sensing system. We then design a supervised learning algorithm, based on a
CNN architecture, for damage detection and severity ranking that uses the measured time series of the sectional aerodynamic
pressure distribution as input. The proposed multivariate algorithm achieves a mean classification accuracy of $91.6\%$ for the
$0°$ AoA dataset and of $89.2\%$ for the $8°$ AoA dataset, when averaged over the three considered splits of the respective datasets.
Furthermore, we determine with reference acceleration measurements and a FE-model of the cantilever beam that the cantilever
beam exhibits complex dynamics, with the resulting measured response not revealing purely monotonic trends (e.g. in terms
of amplitude increase or frequency shifts), which renders the damage identification problem non-trivial.

Although the two datasets of aerodynamic pressure measurements comprise consequently different wind velocities $V$, heaving
frequencies $f_h$, and complex dynamic behavior of the structure, our proposed deep learning based method with only slightly
optimized hyperparameters yields good results, with high classification accuracy and narrow banded confusion matrices for
both datasets and all regarded splits. Thus, we conclude that our initial hypothesis that structural damage can be detected and
rated based on sectional aerodynamic pressure measurements within a mildly turbulent environment and a fixed-wing setup is
indeed valid and that damage detection using aerodynamic pressure measurements from MEMS-based sensing is a path that
warrants further investigation for real-world SHM.

For future research on the use of aerodynamic pressure measurements for damage detection, we consider it essential to better
understand the transient aerodynamic phenomena caused by increased heaving or pitching amplitudes. A deeper understand-
ing of these phenomena appears to be crucial for detecting damage to wind turbine blades in real-world environmental and
operational settings. In addition, we recommend performing similar experiments with a crack orientation that corresponds to
a bending damage and under more realistic operational and environmental conditions. Moreover, we also suggest analyzing
aerodynamic pressure data with more interpretable data analysis methods, since explainable results and damage indicators are
essential for structural health monitoring and subsequent decision making.

*Data availability.* The data of our experiments is available in the following repository: https://doi.org/10.34808/gq12-wx33

*Code availability.* The code being developed within this publication is available on GitHub: https://github.com/phfranz/aerosense_damage_
assessment





*Author contributions.* PF: methodology, investigation, data analysis, software, visualization, writing: original draft, writing: review and editing. IA: conceptualization, design of experiments and wind tunnel set-up, conduction of experiments, methodology, supervision, writing: theory supporting hypothesis (Section 2.1), writing: review and editing. GD: conceptualization, design of experiments and wind tunnel set-up, conduction of experiments, methodology, data analysis, software, supervision, writing: review and editing. JD: conceptualization, wind-tunnel set-up and experiments, methodology, software, supervision, writing: review and editing. AJ: Finite-Element modeling, writing: description of Finite-Element model. AP: methodology, supervision, resources, writing: review and editing. SB: Aerosense project lead, conceptualization, resources, writing: review and editing. EC: conceptualization, methodology, software, resources, supervision, writing: review and editing.

*Acknowledgements.* This work is funded by the BRIDGE Discovery Programme of the Swiss National Science Foundation and Innosuisse (Project Number 40B2-0_187087). The authors acknowledge and thank Prof. Karen Mulleners, head of the Unsteady Flow Diagnostics Laboratory (UNFoLD), from EPFL for providing the wind tunnel and supporting our experiments. A.P. acknowledges support by dtec.bw – Digitalization and Technology Research Center of the Bundeswehr (project RISK.twin). dtec.bw is funded by the European Union – Next GenerationEU.

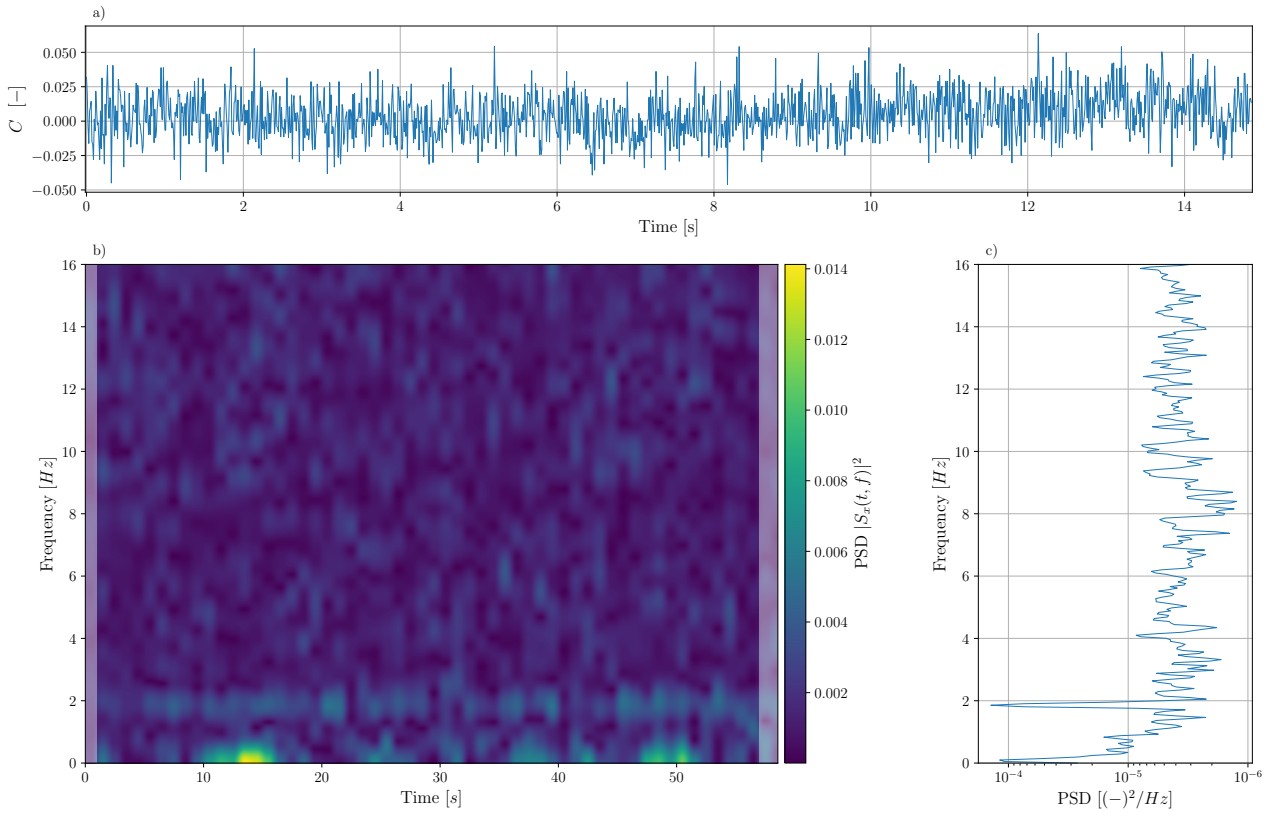

**Figure A1.** a) Extract of the pressure time history recorded by sensor 14, located at $\frac{x}{c} = 0.02$ on the suction side of the airfoil, with $8°$ AoA under $V = 24\frac{m}{s}$ and $f_h = 1.9$Hz in experiment 16. Panel b) shows how the spectrum of the pressure time history of sensor 14 of experiment 16 with $8°$ AoA varies over the whole duration of the experiment via a spectrogram. The spectrogram is computed using the discrete short-term-Fourier-transformation of Scipy (Virtanen et al., 2020). The shaded areas at the left and right ends of the spectrum indicate areas affected by sliding windows that are partially outside of the analyzed signal Panel c) depicts the PSD of the same signal computed over the whole duration of experiment 16 and is calculated with Welch's method and its implementation in Scipy (Virtanen et al., 2020).





**Table A1.** Overview over experiments and their boundary conditions. The entries in column 1, 2 and 3 are the identifiers of the measurements conducted under the boundary conditions of the same row. This numbering is applied for both, the $0°$ and $8°$ dataset.

| V $[\frac{m}{s}]$ | $f_h$ [Hz] | Crack length [% of beam width] | Column 1 | Column 2 | Column 3 |
|---|---|---|---|---|---|
| 12 | 1.0 | 0 | 3 | 4 | 5 |
| 24 | 1.0 | 0 | 7 | 8 | 9 |
| 12 | 1.9 | 0 | 12 | 13 | 14 |
| 24 | 1.9 | 0 | 16 | 17 | 18 |
| 12 | 1.0 | 0 + added mass | 22 | 23 | 24 |
| 24 | 1.0 | 0 + added mass | 26 | 27 | 28 |
| 12 | 1.9 | 0 + added mass | 31 | 32 | 33 |
| 24 | 1.9 | 0 + added mass | 35 | 36 | 37 |
| 12 | 1.0 | 12.5 | 41 | 42 | 43 |
| 24 | 1.0 | 12.5 | 45 | 46 | 47 |
| 12 | 1.9 | 12.5 | 50 | 51 | 52 |
| 24 | 1.9 | 12.5 | 54 | 55 | 56 |
| 12 | 1.0 | 25 | 60 | 61 | 62 |
| 24 | 1.0 | 25 | 64 | 65 | 66 |
| 12 | 1.9 | 25 | 69 | 70 | 71 |
| 24 | 1.9 | 25 | 73 | 74 | 75 |
| 12 | 1.0 | 37.5 | 79 | 80 | 81 |
| 24 | 1.0 | 37.5 | 83 | 84 | 85 |
| 12 | 1.9 | 37.5 | 88 | 89 | 90 |
| 24 | 1.9 | 37.5 | 92 | 93 | 94 |
| 12 | 1.0 | 50 | 98 | 99 | 100 |
| 24 | 1.0 | 50 | 102 | 103 | 104 |
| 12 | 1.9 | 50 | 107 | 108 | 109 |
| 24 | 1.9 | 50 | 111 | 112 | 113 |





**Table A2.** Overview over experiments where only harmonic forces are acting on the cantilever beam. This numbering is applied for both, the $0°$ and $8°$ dataset.

| $f_h$ [Hz] | Crack length [% of beam width] | Number of experiment |
|---|---|---|
| 1.0 | 0 | 2 |
| 1.9 | 0 | 11 |
| 1.0 | 0 + added mass | 21 |
| 1.9 | 0 + added mass | 30 |
| 1.0 | 12.5 | 40 |
| 1.9 | 12.5 | 49 |
| 1.0 | 25 | 59 |
| 1.9 | 25 | 68 |
| 1.0 | 37.5 | 78 |
| 1.9 | 37.5 | 87 |
| 1.0 | 50 | 97 |
| 1.9 | 50 | 106 |

**Table A3.** Eigenfrequencies $f_{i,m}$ of the first five eigenmodes $\phi_{i,m}$ of the FE-model, determined via eigenvalue analysis.

| Crack length [% of the beam width] | eigenmode 1 | eigenmode 2 | eigenmode 3 | eigenmode 4 | eigenmode 5 |
|---|---|---|---|---|---|
| 0 % | 1.929 | 7.682 | 11.906 | 47.408 | 52.070 |
| 12.5 % | 1.920 | 7.650 | 11.896 | 47.288 | 52.053 |
| 25 % | 1.914 | 7.530 | 11.870 | 46.842 | 52.008 |
| 37.5 % | 1.903 | 7.300 | 11.829 | 46.045 | 51.936 |
| 50 % | 1.887 | 6.911 | 11.769 | 44.836 | 51.834 |



**Table A4.** Eigenfrequencies $f_i$ of the vertically oscillating eigenmodes of the cantilever beam below 60Hz for all structural states. $\mu_{f,i}$ and $\sigma_{f,i}$ are computed from the stable model orders of the respective experiment.

| Experiment # | $\phi_1$ | | $\phi_2$ | | $\phi_3$ | | $\phi_4$ | | $\phi_5$ | | $\phi_6$ | |
|---|---|---|---|---|---|---|---|---|---|---|---|---|
| | $\mu_{f,1}$ [Hz] | $\sigma_{f,1}$ [Hz] | $\mu_{f,2}$ [Hz] | $\sigma_{f,2}$ [Hz] | $\mu_{f,3}$ [Hz] | $\sigma_{f,3}$ [Hz] | $\mu_{f,4}$ [Hz] | $\sigma_{f,4}$ [Hz] | $\mu_{f,5}$ [Hz] | $\sigma_{f,5}$ [Hz] | $\mu_{f,6}$ [Hz] | $\sigma_{f,6}$ [Hz] |
| 6 (0) | 1.949 | 0.003 | 13.883 | 0.004 | 23.223 | 0.038 | - | - | 44.932 | 0.002 | 53.233 | 0.077 |
| 15 (0) | 1.951 | 0.008 | 13.894 | 0.013 | 23.239 | 0.031 | - | - | 44.941 | 0.012 | 53.144 | 0.224 |
| 20 (1) | 1.879 | 0.007 | 13.757 | 0.003 | 23.168 | 0.019 | - | - | 42.133 | 0.107 | - | - |
| 25 (1) | 1.883 | 0.002 | 13.739 | 0.002 | 22.715 | 0.019 | - | - | 41.774 | 0.063 | 53.931 | 0.032 |
| 44 (2) | 1.953 | 0.001 | 13.844 | 0.005 | 24.513 | 0.062 | - | - | 44.967 | 0.009 | 52.100 | 0.096 |
| 53 (2) | 1.947 | 0.002 | 13.825 | 0.006 | 24.197 | 0.010 | - | - | 45.006 | 0.001 | 52.411 | 0.121 |
| 63 (3) | 1.942 | 0.002 | 13.790 | 0.003 | 24.663 | 0.018 | - | - | 44.964 | 0.035 | - | - |
| 67 (3) | 1.942 | 0.002 | 13.805 | 0.007 | 24.603 | 0.046 | 34.223 | 0.111 | 45.036 | 0.008 | - | - |
| 82 (4) | 1.929 | 0.001 | 13.788 | 0.002 | 24.676 | 0.020 | 34.279 | 0.053 | 44.963 | 0.004 | 52.635 | 0.294 |
| 86 (4) | 1.919 | 0.008 | 13.791 | 0.005 | 24.618 | 0.046 | 34.209 | 0.014 | 44.834 | 0.001 | 52.767 | 0.294 |
| 101 (5) | 1.906 | 0.002 | 13.723 | 0.002 | 24.897 | 0.026 | 34.229 | 0.050 | 44.825 | 0.003 | 51.286 | 0.107 |
| 111 (5) | 1.901 | 0.003 | 13.729 | 0.002 | 24.623 | 0.063 | 34.034 | 0.053 | 44.802 | 0.002 | 51.801 | 0.351 |



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
