# Peer review of "On the Potential of Aerodynamic Pressure Measurements for Structural Damage Detection"

_Wind Energy Science, 2025_

## Author Comment (AC1)

**wes-2025-26 Response to Anonymous Referee #1**

Thank you for taking the time to review our manuscript. We sincerely appreciate your constructive feedback and careful evaluation. In response, we have revised the manuscript accordingly, with all changes clearly marked in blue for your convenience. Corresponding updates are also indicated in blue font within this response letter. We trust that these revisions address your comments and enhance the clarity and quality of our work, and we hope the updated version meets your expectations.

Key concerns:

1. Material and Damage Representation: The structural damage was introduced in metal rather than in composite materials, which are more representative of real-world turbine blades. Furthermore, a saw cut does not replicate the characteristics of a crack as it would naturally occur.

**Reply:** We thank the reviewer for this valuable observation. We fully acknowledge that our experimental setup, specifically, the use of an aluminum cantilever and the introduction of damage via saw cuts, does not replicate the material composition or crack morphology typical of modern composite wind turbine blades.

Our decision to use a metallic cantilever was driven by the need for a controlled and repeatable environment to establish a proof of concept. The use of a saw cut enables us to systematically vary the damage severity and assess its measurable impact on the aerodynamic pressure field under well-defined and reproducible conditions. While the induced damage may not mimic natural crack propagation mechanisms in composites, it introduces a local stiffness reduction that suffices to validate the hypothesis that structural changes manifest as measurable perturbations in the aerodynamic pressure distribution.

This study is intended as a first step to demonstrate the feasibility of using aerodynamic pressure measurements for structural condition assessment. Future work will focus on extending the methodology to composite specimens and more realistic damage types, such as delaminations or matrix cracking.

We have modified the manuscript in several locations and added a dedicated paragraph to the revised manuscript explicitly acknowledging these limitations and clarifying the motivation behind our experimental choices:

- line 160: We approximate a fixed WTB by mounting an airfoil on an aluminum cantilever beam with a rectangular cross-section and placing it in a wind tunnel test section.
- lines 173-178: An important limitation of our setup lies in the choice of material and the manner in which damage is introduced. While real-world wind turbine blades are composed of layered composite materials exhibiting complex failure modes such as delamination and fiber breakage, our experiments employ an aluminum cantilever with damage emulated via saw cuts. This simplification allows for controlled, repeatable tests and the ability to systematically vary damage severity. Although the artificial crack does not fully replicate the morphology or fracture mechanics of a naturally occurring defect in composites, it produces a measurable stiffness reduction, which is central to our proof-of-concept study.

• lines 589-592: Additionally, more advanced material models and realistic damage representations will be necessary to accurately account for variations in material properties and structural integrity. Future research efforts will aim to translate the proposed methodology to composite specimens to more closely align with practical applications in wind turbine blade monitoring.

Additionally, we sketch in the outlook of the revised manuscript a strategy, how the proposed measurement and detection concept could be extended towards real-world applicability (see lines 583-607):

Beside these immediate next steps, scaling the damage detection approach proposed in this study to full-scale wind turbines and real-world environmental and operating conditions (EOCs) requires substantial further research and development. We propose the following multi-stage strategy to facilitate this scaling process:

- Simulation and experimental validation: Further numerical simulations and experimental validation—such as wind tunnel testing using a miniature wind turbine under varying wind speed conditions and turbulence intensities—are essential. These efforts aim to deepen our understanding of how realistic inflow conditions, rotational aerodynamic effects, and real-world damage scenarios influence the pressure distribution along turbine blades. Additionally, more advanced material models and realistic damage representations will be necessary to accurately account for variations in material properties and structural integrity. Future research efforts will aim to translate the proposed methodology to composite specimens to more closely align with practical applications in wind turbine blade monitoring.
- **Spatial distribution of pressure sensors:** Scaling damage detection to cover the full blade span can be realized by deploying multiple Aerosense sensor nodes along each blade, as illustrated in Figure 1(a) of the manuscript. This setup enables the simultaneous acquisition and processing of aerodynamic pressure data at several chord- and span-wise locations, thereby enhancing spatial resolution and detection capability.
- Unsupervised or self-supervised damage detection: In real-world applications, labeled data are typically unavailable. Therefore, an unsupervised or selfsupervised approach to anomaly or damage detection is required. In ongoing work, we are developing such a method tailored to the dataset presented in this study, to be reported in a forthcoming publication. To adapt this to operational and environmental variability, we propose leveraging local inflow information estimated via other Aerosense methods (see Section 4 of [2] and Section 4.5 of [8]). Moreover, fusing aerodynamic pressure data with measurements from the 6-DOF inertial measurement unit embedded in each Aerosense node may further enhance robustness and sensitivity.
- Field deployment and scaling: The final step involves implementing the proposed sensor layout and detection methods on a small-scale operational wind turbine. Field testing will serve to validate the performance of the unsupervised detection framework. The knowledge and insights gained through this process will inform the subsequent upscaling to full-scale wind turbines, enabling robust aerodynamic pressure-based damage detection under realistic conditions.

According to the third bullet point from above, we add reference [8] to the manuscript.

2. Experimental Conditions: Both the wind excitation and imbalance excitation were kept constant throughout the experiments, a condition not reflective of the variable nature of real-world wind turbine environments.

**Reply:** We appreciate the reviewer's thoughtful comment. We fully agree that the experimental conditions employed, namely constant wind inflow and harmonic excitation, do not capture the full complexity of real-world wind turbine operation.

However, this study was designed as a proof of concept to assess whether structural damage can be reliably detected from aerodynamic pressure measurements in a simplified and well-controlled setting. Given the indirect nature of aerodynamic pressure as a proxy for structural condition, a controlled environment was necessary to observe and isolate the underlying mechanisms governing damage detectability.

Moreover, we acknowledge that our findings are not directly transferable to full-scale wind turbines. Nonetheless, they lay the groundwork for future research and scaled-up experimental campaigns under more realistic and variable operating conditions. We have revised manuscript at several locations to more clearly articulate this scope:

- line 6: This proof of concept study is based on a series of wind tunnel experiments on a NACA 633418 airfoil.
- lines 180-188: Our investigation is conducted in a wind tunnel facility under controlled environmental and operational conditions (EOCs) which do not reflect the complexity of the EOCs real world wind turbines. However, given the indirect nature of aerodynamic pressure as a proxy for structural condition, a controlled environment is necessary to observe and isolate the underlying mechanisms governing damage detectability. Thus, this paper does not aim to answer whether it is possible to detect and rank the severity of structural damage under real operational (rotating wing aerodynamics, pitching, tension stiffening, etc.) and environmental conditions (high turbulence, varying temperature and weather) of a wind turbine and its findings are not directly transferable to full-scale wind turbines. Instead, this paper rather aims to offer a proof of concept as to whether such highly indirect pressure measurements can be conceived for use within an SHM setting and to lay the groundwork for further research and the justification for scaled-up experimental campaigns under more realistic and variable EOCs.
- lines 586-589: These line point to the first bullet point of our scaling strategy in response to key concern 1. Therefore, please find this above.
- 3. Evaluation Methodology: A supervised classification method based on CNNs was employed. For real applications, datasets typically do not include labeled damage states, necessitating unsupervised methods that do not rely on such data.

**Reply**: Thank you for this insightful observation. We fully agree that in real-world applications, labeled data corresponding to specific damage states are generally not available. As a result, supervised classification methods are not directly applicable, and unsupervised or self-supervised anomaly detection methods must be pursued for practical deployment.

In this study, we deliberately begin with the simpler, supervised classification setting to assess the fundamental viability of using indirect aerodynamic pressure measurements for structural damage detection. This controlled setup allows us to verify whether the pressure signals—despite the complexity of the underlying aeroelastic dynamics—indeed

carry identifiable signatures of damage severity. Demonstrating success in this supervised task thus provides a crucial first step in validating the information content and relevance of the measurement modality itself.

Moreover, the findings from this supervised task offer valuable architectural and signalprocessing insights that will inform the development of unsupervised approaches. For instance, the convolutional neural network (CNN) architecture used here may serve as a robust encoder in a future autoencoder-based anomaly detection framework. We are actively working on this transition and will present the results in a forthcoming follow-up publication.

To highlight this rationale and future direction in the manuscript, we have added the following clarifying statements:

- lines 392-395: Although the supervised learning approach employed in this study is not directly applicable to real-world scenarios due to the absence of labeled data, it serves as a first step to assess the viability of indirect aerodynamic pressure measurements for damage detection. The CNN architecture developed here may also provide a suitable encoder for future unsupervised anomaly detection approaches.
- lines 597-603: These lines point to the third point of our scaling strategy in response to key concern 1. As outlined there, future work will focus on the development of unsupervised and self-supervised anomaly detection methods that are suitable for real-world deployment.
- 4. Given these points, the manuscript's findings have limited transferability to real-world settings. Furthermore, the impact and scope of the work may be misaligned with the target journal's focus.

**Reply:** We appreciate your comment and fully acknowledge that our experimental setup does not capture the full complexity of real-world operational conditions. However, we respectfully argue that the value of our study lies in its role as a foundational proof-of-concept. Conducting the investigation under controlled and simplified conditions was an intentional and necessary first step to rigorously evaluate our central hypothesis: that structural damage induces measurable perturbations in the aerodynamic pressure distribution.

This controlled framework allows us to isolate key mechanisms governing damage detectability, validate the efficacy of our indirect sensing approach, and assess the potential of data-driven detection models before transitioning to more complex and variable realworld environments. Such early-stage, hypothesis-driven studies are critical in establishing scientific feasibility and reducing risks in the subsequent development of scalable monitoring systems.

In this sense, while the study may be limited in direct transferability, it contributes meaningful insight to the wind energy and structural health monitoring communities by advancing the understanding of aerodynamic sensing as a viable pathway for damage detection. We believe this aligns with the journal's scope, which includes the development of novel methods with potential for future real-world application.

Specific points for improvement include:

1. Line 281 mentions that the parameters of the CNNs are fewer compared to alternative methods. It would be helpful to specify the number of parameters used in the CNNs.

**Reply** Thank you for this helpful suggestion. We have now included the exact number of trainable parameters used in our CNN model in the revised manuscript (lines 294-296) to provide greater clarity and transparency.

Our decision is motivated by the fact that the number of trainable parameters in the proposed CNN architecture—302,342 parameters—is substantially lower than in the above mentioned alternative architectures, which typically involve significantly more complex networks.

2. Line 284 notes the use of the Adam algorithm. Given that AdamW is now the standard, why was the Adam algorithm chosen?

**Reply**: Thank you for this thoughtful comment. We were not aware of the advantages of AdamW over Adam and compared our proposed CNN against a CNN with the same architecture and training routine using AdamW and a small weight decay of 0.0001. However, the overall classification accuracy yielded via the AdamW model was lower compared to the classification accuracy reported in the preprint, as can be seen in Table 1. The table reports slightly lower classification accuracy for both datasets.

Table 1: Classification results using the proposed CNN architecture and training procedure together with the AdamW optimizer and a weight decay of 0.0001.

| AoA | split $1$ | split $2$ | split $3$ | average |
|-----|-----------|-----------|-----------|---------|
| 0°  | 80.00%    | 97.80%    | 94.66%    | 90.82%  |
| 8°  | 79.40%    | 94.43%    | 87.12%    | 86.98%  |

To increase the classification accuracy, we introduced - further to the weight decay implemented by AdamW - additional regularization in the form of standard dropout layers after the activation of every convolutional layer.1 For a dropout rate of 0.2 and a weight decay of 0.0001, we obtain the accuracies on the different splits shown in Table 2:

Table 2: Classification results using the proposed CNN architecture enhanced by dropout layers (dropout rate of 20%) after every convolutional layer and proposed training procedure together with the AdamW optimizer and a weight decay of 0.0001.

| AoA | split 1 | split 2 | split 3 | average |
|-----|---------|---------|---------|---------|
| 0°  | 82.77%  | 98.70%  | 95.10%  | 92.19%  |
| 8°  | 82.63%  | 95.70%  | 88.20%  | 88.20%  |

The accuracy on the different splits shown in Table 2 is similar to the accuracy of the model proposed in the preprint, for each single split and thus also for the average values. Training either the AdamW or the AdamW + dropout model for more epochs, using a different learning rate scheme, for example cosine annealing or exponential decay, or varying the weight decay didn't increase the accuracy of the model.

Furthermore, in response to your remark 5, we compare the loss curves of our proposed model from the preprint with those of the AdamW + dropout variant, as both achieve

<sup>1This model is subsequently referred to as 'AdamW + dropout' model

Figure 1: Training loss curve, validation loss curve and the curve of the stepwise decreasing learning rate of the model suggested in the preprint, plotted over the training epochs. These curves correspond to training on split 2 of the 0° AoA dataset.

similar accuracy across all data splits. Figure 1 presents the training loss, validation loss, and learning rate curves for the model proposed in the preprint. While the training loss decreases steadily with only minor fluctuations, the validation loss exhibits pronounced oscillations during the early training phase—particularly when the learning rate is high. As the learning rate progressively decreases, the magnitude of these fluctuations in the validation loss also diminishes. In the final  $\sim 40$  iterations, the validation loss stabilizes and fluctuates only mildly. Overall, the model demonstrates good convergence behavior, despite the initial variability in the validation loss.

Figure 2: Training loss curve, validation loss curve and the curve of the stepwise decreasing learning rate of the AdamW and dropout model described above, plotted over the training epochs. These curves correspond to training on split 2 of the 0° AoA dataset.

Figure 2 shows the training loss, validation loss, and learning rate curves for the AdamW + dropout model. Similar to the model proposed in the preprint, the training loss decreases consistently over time (see panel a) of Figure 3). However, the validation loss exhibits more pronounced fluctuations during the final ~50 training epochs when compared to the validation loss of the preprint model (see panel b) of Figure 3). Additionally, the learning rate does not decay as substantially as it does in the preprint model (again, see panel a) of Figure 3).

When examining the smoothed validation loss curves in panel b)—of Figure 3 obtained using debiased exponential smoothing with a smoothing factor of 0.7 (as implemented in TensorBoard)—it becomes evident that the preprint model converges toward lower validation loss values. Given this slightly more stable training behavior, the reduced architectural complexity, and the absence of any significant difference in classification accuracy between the two models, we have opted to retain the preprint model in this study.